# Systematic clustering algorithm for chromatin accessibility data and its application to hematopoietic cells

**Azusa Tanaka**[1,2]☯*, **Yasuhiro Ishitsuka**[3,4]☯*, **Hiroki Ohta**[3,5]☯*, **Akihiro Fujimoto**[1], **Jun-ichirou Yasunaga**[2,6], **Masao Matsuoka**[2,6]

**1** Department of Human Genetics, Graduate School of Medicine, The University of Tokyo, Tokyo, Japan, **2** Laboratory of Virus Control, Institute for Frontier Life and Medical Sciences, Kyoto University, Kyoto, Japan, **3** Center for Science Adventure and Collaborative Research Advancement, Graduate School of Science, Kyoto University, Kyoto, Japan, **4** Department of Mathematics, Graduate School of Science, Kyoto University, Kyoto, Japan, **5** Department of Physics, Graduate School of Science, Kyoto University, Kyoto, Japan, **6** Department of Hematology, Rheumatology and Infectious Disease, Faculty of Life Sciences, Kumamoto University, Kumamoto, Japan

☯ These authors contributed equally to this work.
* a-tanaka@m.u-tokyo.ac.jp (AT); yasu-ishi@math.kyoto-u.ac.jp (YI); ohta.hiroki.6c@kyoto-u.ac.jp (HO)

**Data Availability Statement:** All ATAC-seq and RNA-seq data needed to reproduce this study have been deposited at the DNA Data Bank of Japan (DDBJ) under accession number DRA010939. The

## Abstract

The huge amount of data acquired by high-throughput sequencing requires data reduction for effective analysis. Here we give a clustering algorithm for genome-wide open chromatin data using a new data reduction method. This method regards the genome as a string of 1s and 0s based on a set of peaks and calculates the Hamming distances between the strings. This algorithm with the systematically optimized set of peaks enables us to quantitatively evaluate differences between samples of hematopoietic cells and classify cell types, potentially leading to a better understanding of leukemia pathogenesis.

## Author summary

High-throughput sequencing provides us huge amounts of data about gene regulation. In order to extract useful information from the data, data reduction is needed. Although RNA-seq data analysis has been extensively studied, where the focus is mainly on genetic loci, tools for epigenetic sequencing data, such as ATAC-seq data which represent chromatin accessibility, are comparatively lacking. Since the binding of transcription factors mainly occurs in open chromatin regions, it is presumably important to understand how chromatin accessibility landscape affects cell phenotype. In this context, we developed a systematic algorithm to select a set of peaks representing the open state of chromatin for a given sample of ATAC-seq data. This algorithm quantifies the difference between samples by regarding the genome as a string of 1s and 0s with Hamming distances and then performs hierarchical clustering. This algorithm has less computational cost and gives a reasonable cell type classification compared to a previous method. In this work, as an application of this algorithm, we present a comparative analysis of leukemia samples with

source code is available from https://github.com/tanakanishi/findclosest.

**Funding:** This research was supported by JSPS KAKENHI Grant Numbers JP19K16740 (AT), JP18J40119 (AT), JP19H03689 (MM), JP20H03514 (JiY), and by Japan Agency for Medical Research and Development (AMED) Grant Numbers JP20fk0108088h0002 (MM), JP17km0405207h0002 (AF), JP18km0405207S0103 (AF), and by a grant from the Naito Foundation (AT). The funders had no role in study design, data collection and analysis, decision to publish, or preparation of the manuscript.

**Competing interests:** The authors declare that they have no conflict of interest.

healthy hematopoietic cells and provide new insights about the relationship between chromatin structures, cell surface proteins, and symptoms in leukemia.

## Introduction

Cellular phenotypes are governed by epigenetic mechanisms. For example, information about how human DNA is packed and chemically modified in the nucleus plays an important role in understanding the differentiation and regulation of cells [1–4]. Methods such as chromatin immunoprecipitation sequencing (ChIP-seq) and assay for transposase accessible chromatin using sequencing (ATAC-seq) have proven useful for understanding the modification and detection of open chromatin on a genome-wide scale [5–9]. Those epigenetic data analysis methods usually start with data enrichment along the whole genome, also known as "peak calling" [10, 11].

Compared to RNA-seq data analysis, whose target regions are mainly in certain loci or genes across samples, the target regions on epigenetic sequencing data are undetermined. To determine the target regions, peak calling with an appropriate tool is often performed for the entire genome of every sample, and the target regions are defined as merged peaks among all samples. Then the total number of reads or fragments present in each region is counted for each sample, leading to a matrix, $X = (x_{i,j})$, where $x_{i,j}$ represents the number of reads/fragments from sample $i$ in region $j$. The matrix elements are normalized by quantile normalization to reduce the biases arising from variations in the data size over samples, followed by downstream processing [7–9].

However, this process raises two concerns. First, we do not fully understand the effect of merging all the peaks from different samples. For example, if two peaks from different samples slightly overlap, those two peaks are considered as one peak after the peak merging step. Therefore, the difference of the two peak positions, which may reflect cell identity, may be unintentionally ignored. The second concern is that we have no justification for applying quantile normalization over samples that are phenotypically different [12, 13].

Thus, the aim of the present study is to avoid these concerns by constructing an algorithm that systematically classifies epigenetic data obtained from high-throughput sequencing. In this analysis, toward cell type classification, we provide a systematic algorithm to select a set of peaks used for the downstream analysis, where the difference between samples are quantified by using the Hamming distance from information theory [14]. This algorithm has less computational cost while still producing reasonable classification compared to a previous method [7].

As an application of the developed algorithm, we use it to obtain new insights on samples of leukemia cells from chronic lymphocytic leukemia (CLL), acute myeloid leukemia (AML), and adult T-cell leukemia (ATL) at the chromatin level. In particular, using this algorithm, we infer the phenotype of a given leukemia sample as output by using only ATAC-seq data of that sample as input.

## Results

### ATAC-seq samples

In this paper, we mainly focused on 77 ATAC-seq datasets from 13 human primary blood cell types [7] as test data. The 13 cell types are comprised of hematopoietic stem cells (HSC), multipotent progenitor cells (MPP), lymphoid-primed multipotent progenitor cells (LMPP), common myeloid progenitor cells (CMP), megakaryocyte-erythroid progenitor cells (MEP),

**Table 1. Immunophenotypes of samples.** Types of hematopoietic cells and their corresponding cell surface markers in [7]. For example, CD34+ and CD38- for cell type $v$ means that a cell of type $v$ expresses CD34 but not CD38 at its surface.

| Cell type ($v$) | Number of replicates | Immunophenotypes |
|---|---|---|
| HSC | 7 | Lin-, CD34+, CD38-, CD10-, CD90+ |
| MPP | 6 | Lin-, CD34+, CD38-, CD10-, CD90- |
| LMPP | 3 | Lin-, CD34+, CD38-, CD10-, CD45RA+ |
| CMP | 8 | Lin-, CD34+, CD38+, CD10-, CD45RA-, CD123+ |
| MEP | 7 | Lin-, CD34+, CD38+, CD10-, CD45RA-, CD123- |
| GMP | 7 | Lin-, CD34+, CD38+, CD10-, CD45RA+, CD123+ |
| CLP | 5 | Lin-, CD34+, CD38+, CD10+, CD45RA+ |
| NK | 6 | CD56+ |
| B | 4 | CD19+, CD20+ |
| CD4$^+$T | 5 | CD3+, CD4+ |
| CD8$^+$T | 5 | CD3+, CD8+ |
| Mono | 6 | CD14+ |
| Ery | 8 | CD71+, GPA+, CD45-low |

granulocyte-macrophage progenitor cells (GMP), common lymphoid progenitor cells (CLP), natural killer cells (NK), B cells, CD4$^+$T cells (CD4$^+$T), CD8$^+$T cells (CD8$^+$T), monocytes (Mono) and erythroids (Ery). These cell types are experimentally categorized by immunophenotypes described by the combination of cell surface markers shown in Table 1.

For convenience, $\mathbb{T}$ denotes a set of the thirteen cell types;

$$\mathbb{T} = \{B, CD4^+T, CD8^+T, CLP, CMP, Ery, GMP, HSC, LMPP, MEP, Mono, MPP, NK\}.$$

For all 77 samples, we assigned ATAC-seq reads to reference genome hg19 (http://hgdownload.cse.ucsc.edu/goldenPath/hg19/database/), and among them only those which had high mapping quality values (MQ $\geq$ 30) were used for the peak calling by MACS2 (see S1 Appendix for details of the preprocessing) [15]. The peak calling results consisted of the location with a peak width and the associated $p$-value. Concretely, the location of the k-th peak is expressed by $g_k = (\gamma_k, \alpha_k, \beta_k)$, where $\gamma_k$ is the chromosome number, $\alpha_k$ is the start position, and $\beta_k$ is the end position. Note that we used MACS2 to call all ATAC-seq peaks with the following parameters (- -nomodel - -nolambda - -keep-dup all -p $p_G$), where the number of peaks is affected by the peak calling parameter '-p $p_G$'. The parameter $p_G$ is larger than any $p$-values of the peak calling results. (See Materials and methods for details of the peak-calling).

Note that the peak position depends on parameter $p_G$ of the MACS2 algorithm as shown in Fig 1. For example, the start and end positions of a peak could change and one peak could split into two peaks depending on $p_G$. Thus, we need to take into account the dependence of a set of peaks on different values of $p_G$ for careful analysis.

## Parameterized binarization

First we ranked the peak results in the order of ascending $p$-values and then investigated the relationship between the peak width and the corresponding ranking. We found that as the $p$-value increased, the width of the ATAC-seq peaks became shorter statistically, which suggested the feasibility of robust data reduction against small noise in the data by selecting peaks with smaller $p$-values (Fig 2).

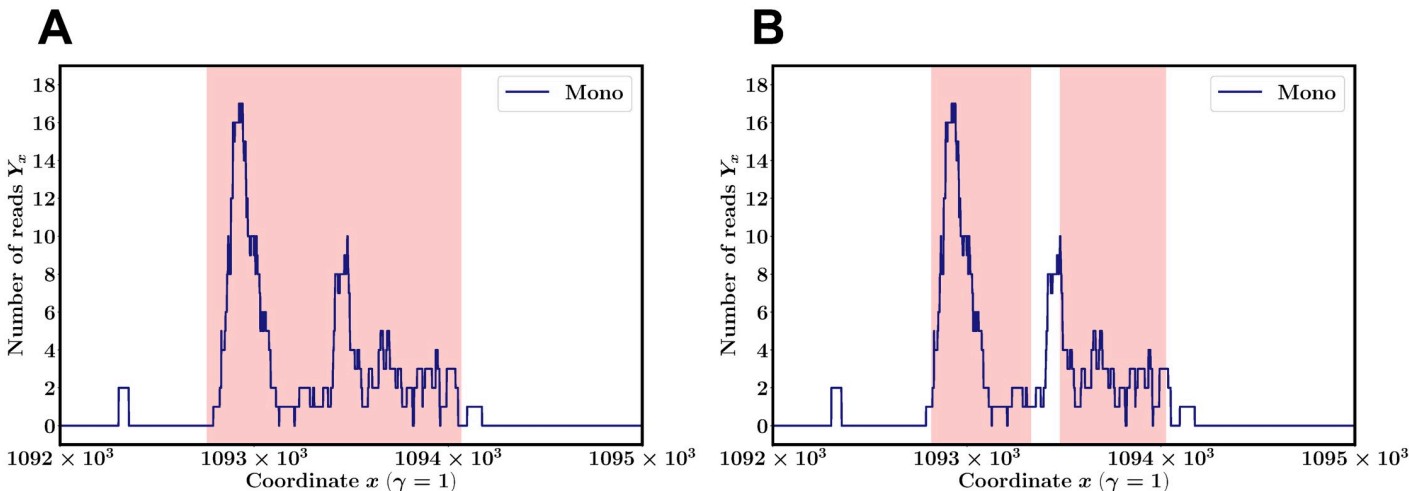

**Fig 1. The number of reads vs genomic positions.** The plots show representative data of Mono obtained from SRA with accession number SRR2920475. (A) The number of reads $Y_x$ at each position $x$ along chr 1 ($\gamma = 1$) and the peak region ($\alpha_k, \beta_k$) as determined by the MACS2 algorithm with peak calling parameter $p_G = 10^{-2}$ (pink shaded regions) is shown. The peak region and its associated $p$-value (($\alpha_k, \beta_k$), $p_k$) are (1092756, 1094068, $10^{-20.36428}$). (B) The obtained peak regions are ((1092817, 1093330), $10^{-20.36428}$) and ((1093480, 1094025), $10^{-8.19447}$) for $p_G = 10^{-4}$.

Thus, we define $M_{\mathrm{cut}}$ as the threshold such that only peaks with rankings not greater than $M_{\mathrm{cut}}$ are used for the analysis hereafter. Then, for a given set of ($M_{\mathrm{cut}}, p_G$), we introduce $\mathbf{B} = \{h_{\gamma,x}\}$, where $h_{\gamma,x} = 1$ when position $x$ in chromosome $\gamma$ is inside a peak and 0 otherwise (Fig 3). The process to obtain the binary sequence from the reads data is illustrated in Fig 4. Note that we do not perform any coarse-grained description for the genome position $x$ but keep 1bp resolution. (See Materials and methods for details of the binarization).

## Quantifying differences between two binary sequences by Hamming distance

Let us move onto the situation when one considers a set of samples to evaluate the difference between two binary sequences $\mathbf{B}$. Here our strategy is to find the proper distance that can be measured from the normalized ATAC-seq data of two samples. Using that distance, we try to obtain hierarchical clustering of a set of hematopoietic cell samples to quantitatively characterize the relationship among those samples.

Let $N_s$ be the number of samples. We then write the set of samples as

$$\mathbb{S} := \{1, 2, \ldots, N_s\},$$

where $N_s = 77$ in this study. For sample $c \in \mathbb{S}$, we add index $c$ to related objects as a superscript. For example, we write a binary sequence $\mathbf{B}$ associated to sample $c$ as $\mathbf{B}^c := \{h_{\gamma,x}^c\}$.

There are many methods to evaluate the difference between a binary sequence $\mathbf{B}^c$ from sample $c \in \mathbb{S}$ and $\mathbf{B}^{c'}$ from sample $c' \in \mathbb{S}$. In this paper, we evaluated the difference between two samples ($c, c'$) by using the Hamming distance $H(\mathbf{B}^c, \mathbf{B}^{c'})$ between two binary sequences, $\mathbf{B}^c$ and $\mathbf{B}^{c'}$. $H(\mathbf{B}^c, \mathbf{B}^{c'})$ is calculated as the sum of the number of pairs with different values at every position $x$ between $\mathbf{B}^c$ and $\mathbf{B}^{c'}$ (Fig 5). We used the distance as an initial condition for the hierarchical clustering and then used Ward's method to complete the hierarchical clustering [16]. Examples of hierarchical clustering with ($M_{\mathrm{cut}}, p_G$) = (2000, $10^{-2}$) and (80000, $10^{-2}$) are shown

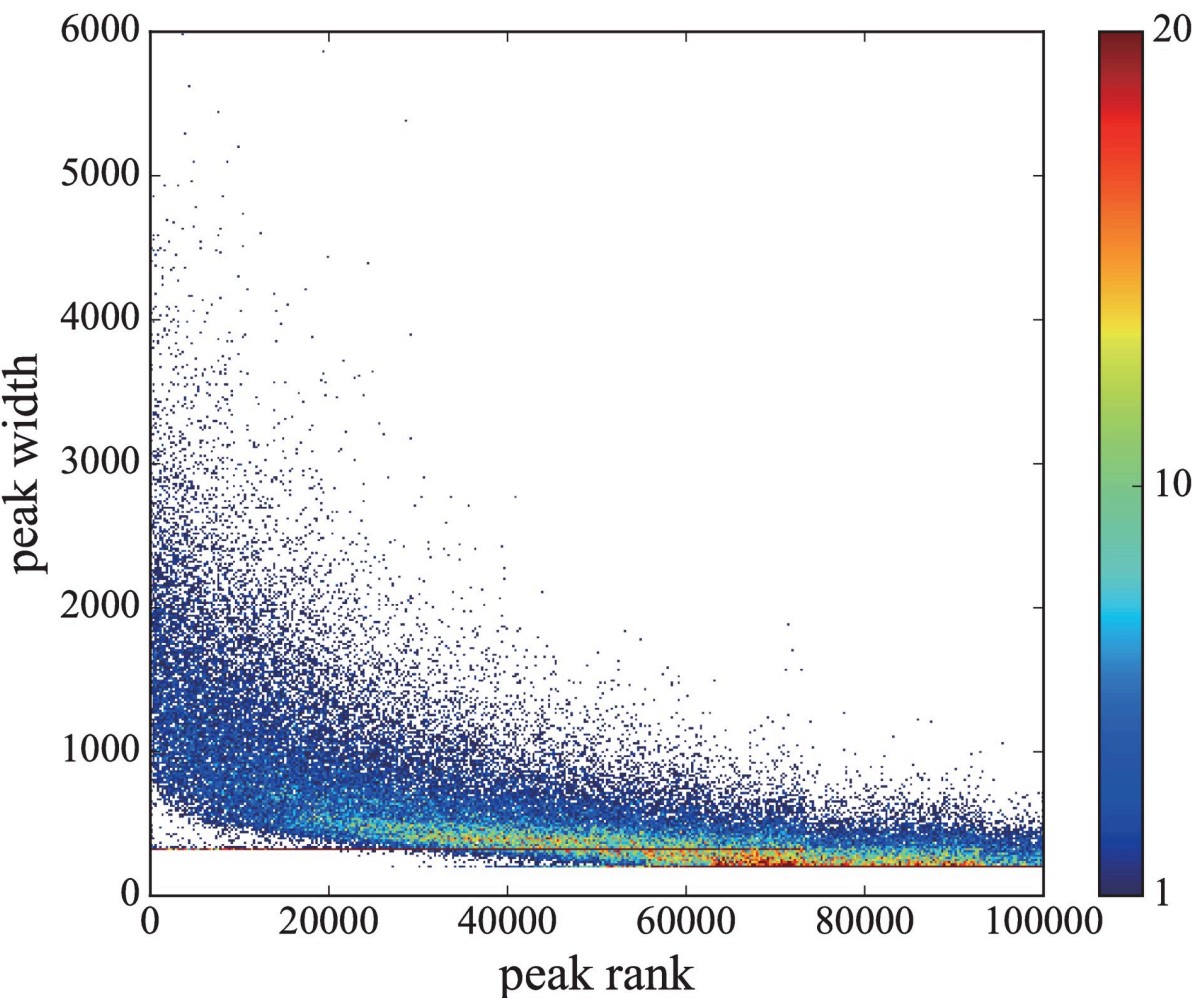

**Fig 2. The statistics of peak width.** Distribution of peak width ($\beta_k - \alpha_k$) and its corresponding ranking $k$ obtained from the peak calling result of CD4$^+$T cells with peak calling parameter $p_G = 10^{-2}$. The bin size is $400 \times 400$. The color code indicates the number of data in each bin.

in Fig 6. (See Materials and methods for details of the Hamming distance and hierarchical clustering).

## Optimization of hierarchical clustering toward cell-type classification

By using the methods explained above, we can obtain a clustering dendrogram that depends on ($M_{\mathrm{cut}}$, $p_G$). We then need to systematically determine the best clustering, which is the clustering closest to the "perfectly classified dendrogram" where each set $\mathbb{S}_v$ of all samples with type $v \in \mathbb{T}$ coincides with an offspring set. This condition can be restated as an optimization problem by introducing a cost function "penalty" for the performance of clustering as follows.

Concretely, to quantitatively evaluate the obtained dendrogram for each combination of ($M_{\mathrm{cut}}$, $p_G$), we define type penalty $\lambda_v$ for a given cell type $v \in \mathbb{T}$. Type penalty $\lambda_v$ corresponds to the number of samples from different cell types in cluster $v$ formed when all samples of cell type $v$ meet together from the bottom of the dendrogram (Fig 7). Additionally, we define

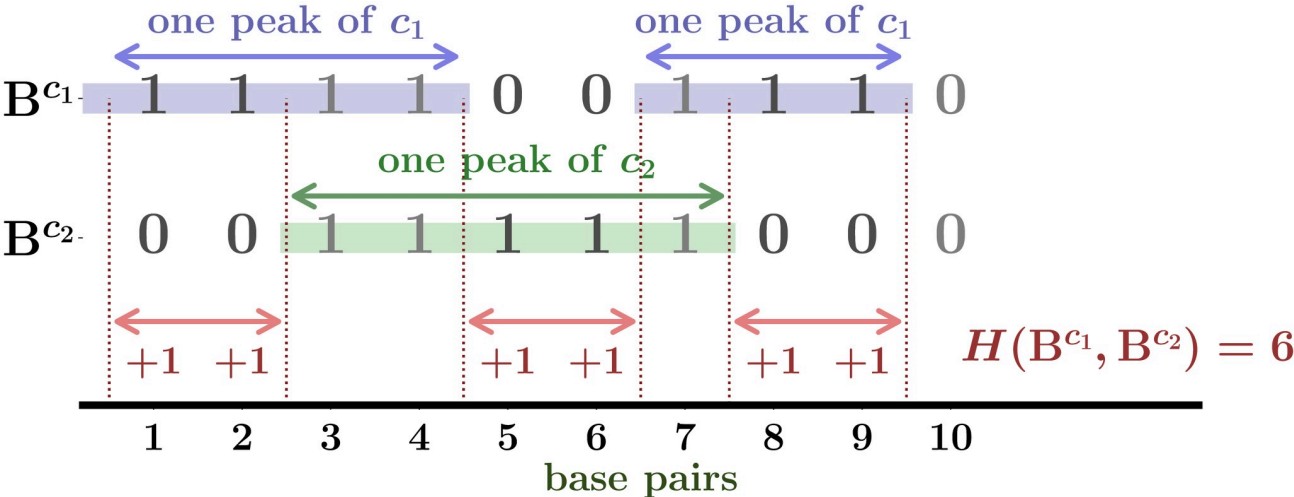

**Fig 3. How to calculate Hamming distance.** Schema of the Hamming distance calculation from the peak locations with two samples $c_1, c_2 \in \mathbb{S}$. Each locus is converted to 1 or 0 based on the peak overlapping status.

global penalty $\lambda := \sum_{v \in \mathbb{T}} \lambda_v$ as the "cost function" of the optimization. Note that $\lambda \geq 0$, and a "perfectly classified dendrogram" gives $\lambda = 0$. (See Materials and methods for details of the penalty).

## Determination of the best parameters for the optimization

As mentioned above, the optimization problem we have to solve is to find $(M_{\text{cut}}^*, p_G^*)$ that minimizes the cost function $\lambda(M_{\text{cut}}, p_G)$. The schematic workflow in our algorithm is shown in Fig 8.

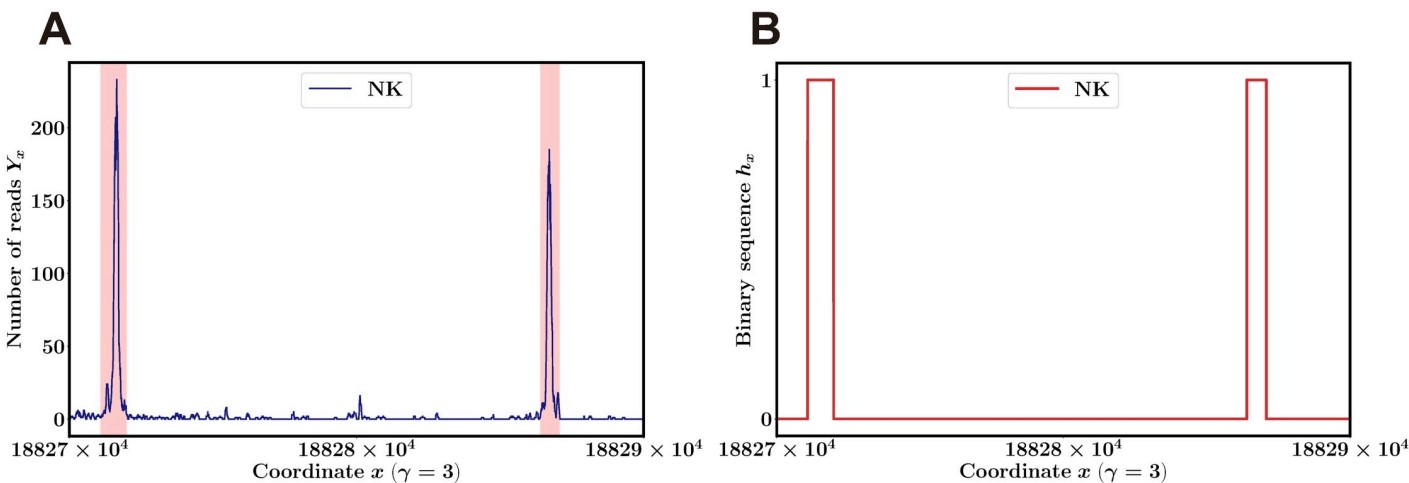

**Fig 4. Binarizing the number of reads.** (A) The number of reads $Y_x$ at each position $x$ along chr 3 ($\gamma = 3$) and the peak region ($\alpha_k, \beta_k$) as determined by the MACS2 algorithm with peak calling parameter $p_G = 10^{-2}$ (pink shaded regions). This figure shows representative data of NK cells obtained from SRA with accession number SRR2920495. The peak regions and the associated $p$-values (($\alpha_k, \beta_k$), $p_k$) in the left and right peaks are ((188271079, 188271985), $10^{-422.5872}$) and ((188286401, 188287077), $10^{-329.52139}$), respectively. Thus, the width of the peaks ($\beta_k - \alpha_k$) in the left- and right-hand sides are 906 and 676, respectively. (B) Binary sequence ($h_x$) as determined by the peak regions seen in (A) when we chose $M_{\text{cut}}$ satisfying $p_{M_{\text{cut}}} \geq 10^{-329.52139}$.

|  | sample 1 | sample 2 | sample 3 | ..... | sample N |
|---|---|---|---|---|---|
| sample 1 | 0 | $d_{12}$ | $d_{13}$ |  | $d_{1N}$ |
| sample 2 | $d_{21}$ | 0 | $d_{23}$ |  | $d_{2N}$ |
| sample 3 | $d_{31}$ | $d_{32}$ | 0 |  | $d_{3N}$ |
| ..... | ..... | ..... | ..... | ..... | ..... |
| ..... | ..... | ..... | ..... | ..... | ..... |
| sample N | $d_{N1}$ | $d_{N2}$ | $d_{N3}$ | ..... | 0 |

**Fig 5. Matrix of Hamming distances.** Matrix of Hamming distances $d_{ij}$ between samples $i$ and $j$. This matrix is used for the downstream analysis.

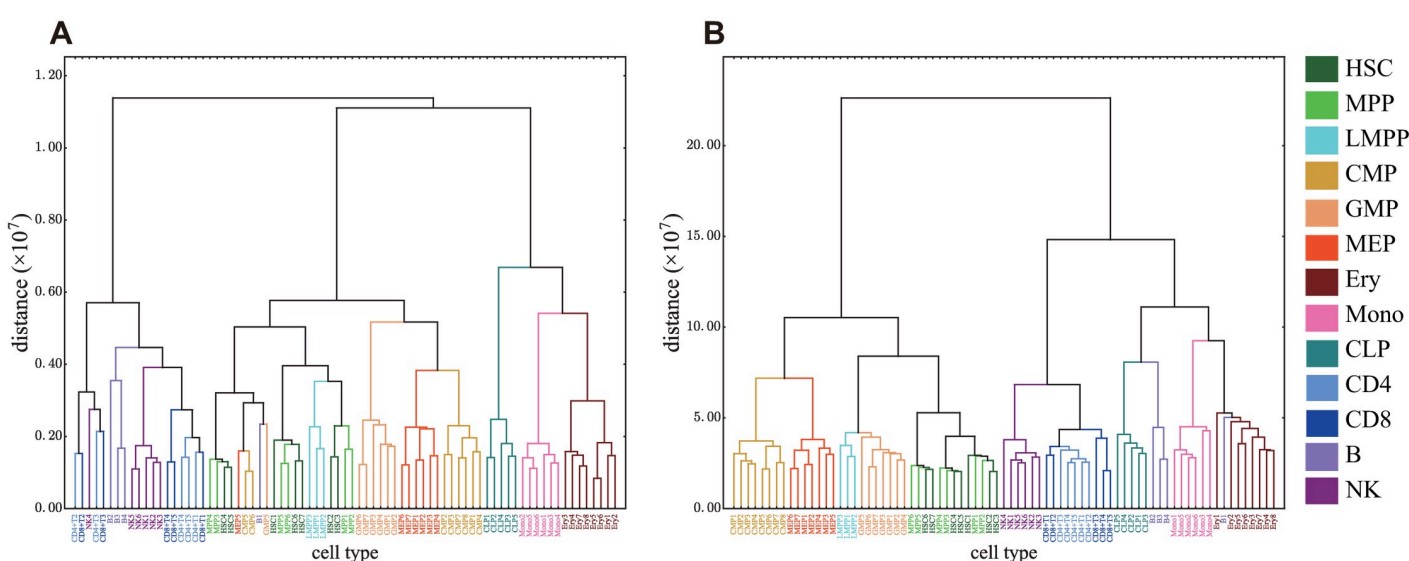

**Fig 6. Examples of clustering dendrograms.** Hierarchical clustering obtained by Ward's method with parameters $(M_{\mathrm{cut}}, p_G) = (2000, 10^{-2})$ (A) and $(80000, 10^{-2})$ (B).

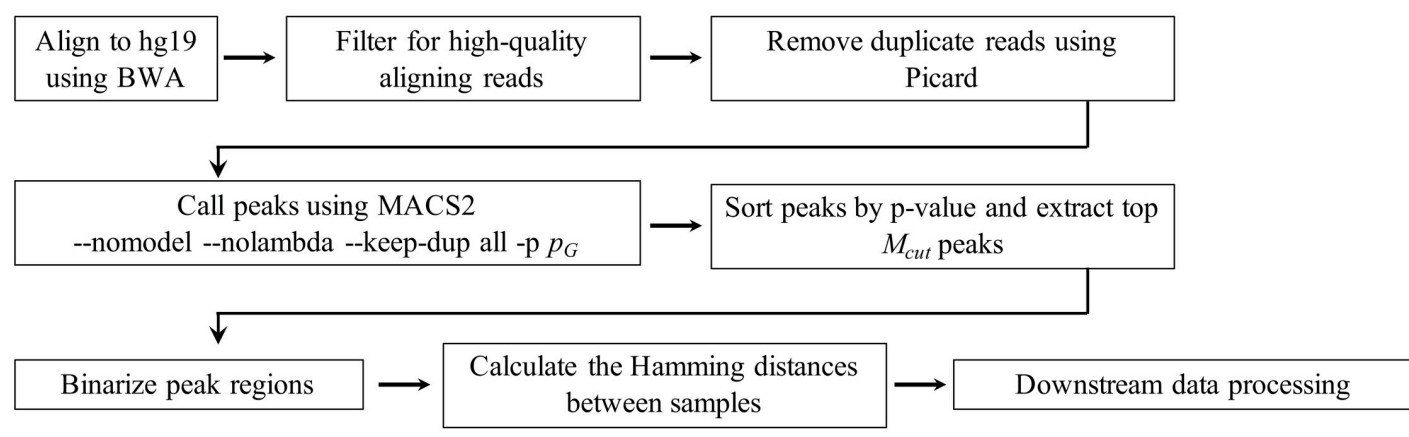

**Fig 7. Schema of penalty score calculation.** Note that this dendrogram is constructed by artificial data to explain how to calculate the penalty, though we use the same labels such as HSC1. This dendrogram has six leaves, and three of them are classified to type HSC. To explain details of this dendrogram, we freely use the symbols and definitions in Materials and methods in this caption. We can see that $\tau(\text{HSC}) = 10$. The corresponding node is $n_{10}$ (displayed by the blue dot), and the corresponding cluster $\mathbb{C}_{10}$ is the set {HSC1, HSC2, HSC3, MPP} (surrounded by the blue dashed line). Among the elements of $\mathbb{C}_{10}$, one leaf, MPP, is not in type HSC, but the three others are. Hence, the type penalty of HSC in this figure is computed as $\lambda_{\text{HSC}} = 4 - 3 = 1$.

**Fig 8. Schematic workflow of our algorithm.** See Materials and methods for details.

First we took into account all the peaks by setting $M_{cut} = \infty$ and checked how the dendrograms and $\lambda(\infty, p_G)$ depended on $p_G$, as shown in Fig 9. Considering the tendency of the parameter searching, we concluded that $1.5 \leq -\log_{10} p_G^* \leq 4$.

We then sought the best parameters to optimize the dendrograms and found that $(M_{cut}^*, p_G^*)$ was close to $(64000, 10^{-2})$, which gave the smallest penalty $\lambda$ in our searching resolution, as shown in Figs 10 and 11. Note that 64000 is the midpoint of (60000, 62000, 64000, 66000, 68000) which give the same minimum penalty in our searching resolution. Hereafter, to investigate the property of the best clustering, we set $(M_{cut}^*, p_G^*)$ as $(64000, 10^{-2})$. In our searching resolution, the increment in terms of $M_{cut}$ was 2000 near $M_{cut} = 64000$. Note that more-refined resolutions might give better estimates of the optimized value $(M_{cut}^*, p_G^*)$, but naturally the computational costs get higher. Even then, the following procedures are operationally unchanged.

The value of the minimum penalty achieved at $(M_{cut}^*, p_G^*)$ was 18. This minimum was smaller than the penalty value of 27 for the clustering of the data from GSE74912_ATACseq_All_Counts.txt in [7]. The procedure of the latter clustering was as follows. First we performed a quantile normalization of the reads count in the distal elements ($> 1000$ bp away from a transcription start site (TSS)). Then we calculated the Pearson coefficients over all samples leading to a distance matrix where each entry is 1-(Pearson coefficient). By using Ward's method, we finally obtained the clustering dendrogram. Note that for this case, Ward's method gives penalty $\lambda = 27$ and UPGMA gives $\lambda = 29$.

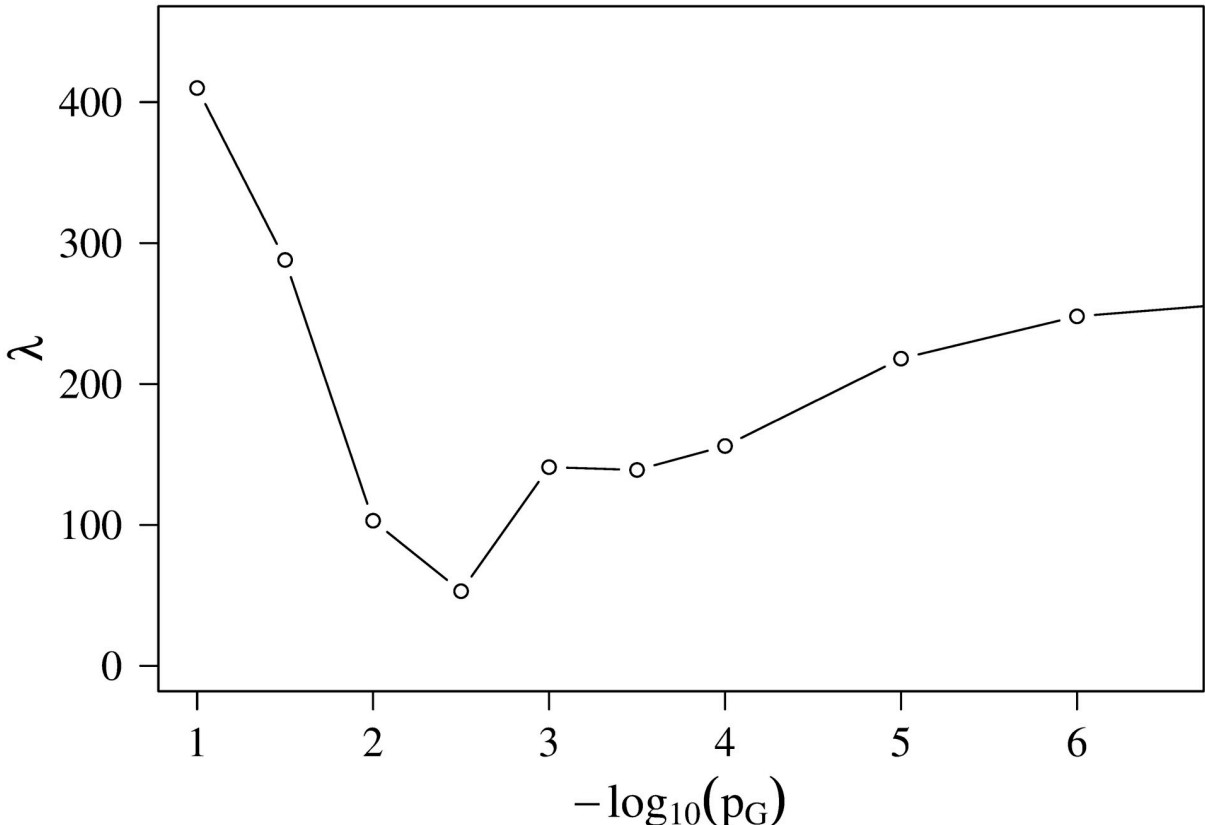

**Fig 9. Global penalty without cutoff of reads.** Global penalty $\lambda(M_{cut} = \infty, p_G)$ obtained by Ward's method.

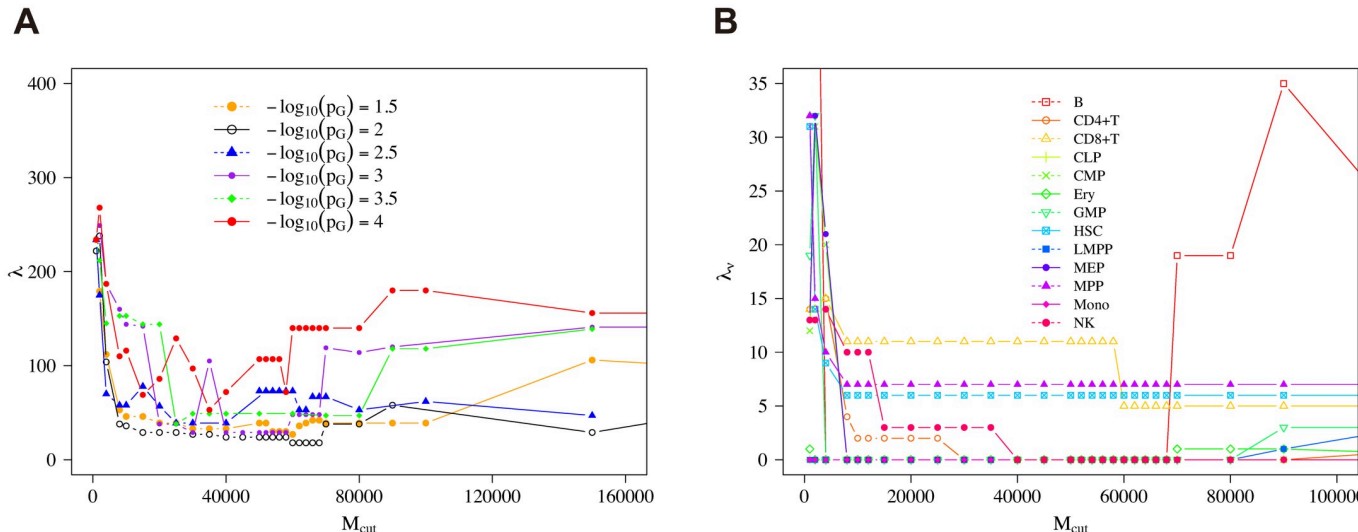

**Fig 10. Penalty with cutoff of reads.** The distribution of global penalty $\lambda$ (A) and type penalty $\lambda_\nu$ for each cell type $\nu$ (B) along with $M_{\text{cut}}$ with parameter $p_G = 10^{-2}$ by Ward's method.

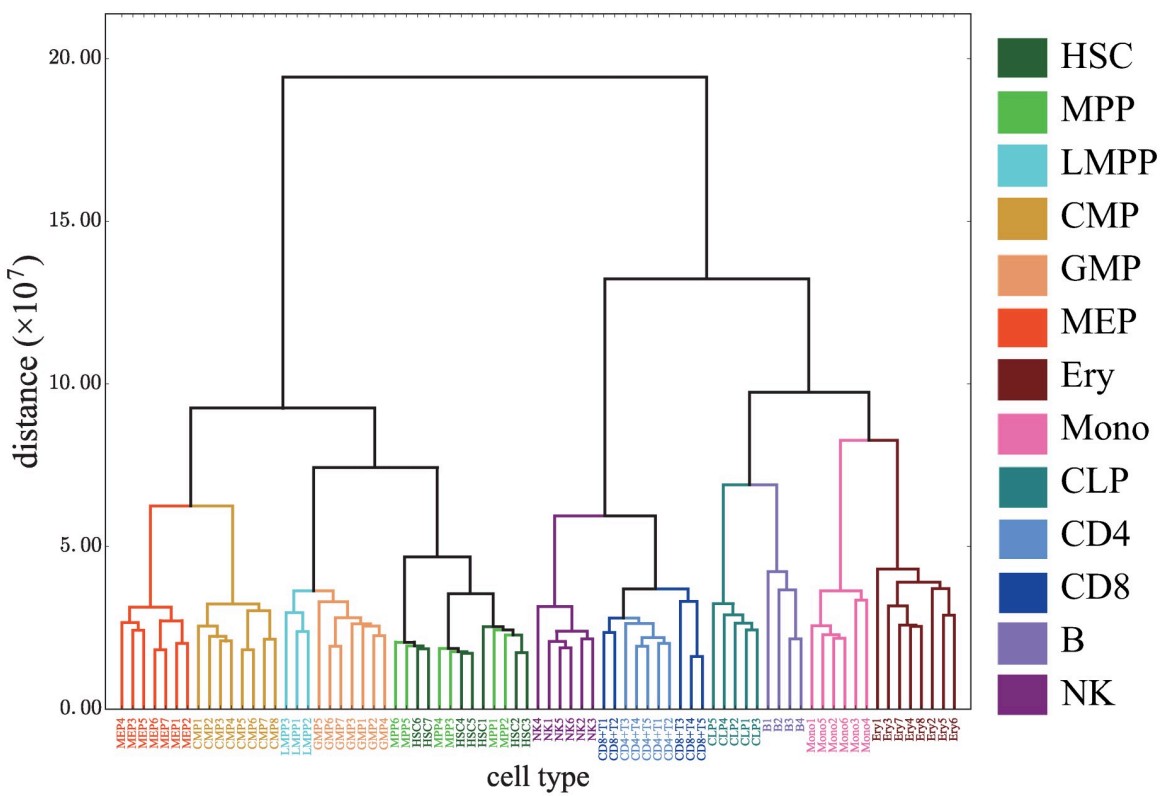

**Fig 11. Our best clustering dendrogram.** Hierarchical clustering obtained by Ward's method with $(M_{\text{cut}}, p_G) = (64000, 10^{-2})$.

## Computational cost of the algorithm

As explained above, after obtaining data of the reads positions, we perform the MACS2 algorithm to get peak regions, and then finally we produce a hierarchical clustering. Here we consider the computational cost of our algorithm after acquiring the data of the reads positions and until acquiring a distance matrix to produce the hierarchical clustering. Note that the computational cost of the MACS2 algorithm is not more than $O(N_s)$, where $O()$ is the Landau notation and $N_s$ is the total number of samples. We consider two situations. (i) One is the case where new samples to analyze are given. (ii) The other is the case where one new sample to analyze is added to the already analyzed samples, for which peak regions and the distance matrix are already calculated. For case (ii), we use the symbol $N_s$ to write the total number of already analyzed samples. We claim that the computational cost of our algorithm is significantly lower than that of a previous method using target regions merged over samples [7] for large values of $N_s$ for case (ii) and, in our case with $N_s = 77$, that the computational cost of our algorithm is practically lower for case (i).

Specifically, in case (i) for our algorithm, the corresponding computational cost is $K_1 M_{\mathrm{cut}} N_s^2$, which comes solely from the calculation of the Hamming distance. In case (ii), the corresponding computational cost is $K_2 M_{\mathrm{cut}} N_s$, which also comes solely from the calculation of the Hamming distance. Note that $K_1$ and $K_2$ are constants that do not depend on $M_{\mathrm{cut}}$ or $N_s$.

In the context of estimating the best optimization parameter $M_{\mathrm{cut}}^*$, by using $M_m$ different values for $M_{\mathrm{cut}}$, the computational cost becomes $K_1 M_{\mathrm{cut}} M_m N_s^2$ for case (i) and $K_2 M_{\mathrm{cut}} M_m N_s$ for case (ii), where $M_m$ does not depend on $N_s$ or genome size $L$ and can be adjusted according to the searching resolution of the optimization. Note that $K_1$ and $K_2$ do not depend on $M_m$. In addition, we optimize $p_G$ by $M_p$ different values for $p_G$. Since this optimization can be done for any algorithm, we do not take into account this cost for the comparison of different algorithms. Typically, we set $(M_m, M_p) \simeq (30, 10)$ in our optimization corresponding to case (i). Note that in the section of "Application to leukemic cells" discussed later, corresponding to case (ii), we use the optimized parameters $(M_{\mathrm{cut}}, p_G) = (M_{\mathrm{cut}}^*, p_G^*)$, leading to $(M_m, M_p) = (1, 1)$.

The previous method using targeted regions merged over samples in [7] includes (a) the merging of reads before peak calling and (b) calculating the distance matrix by the Pearson coefficients which automatically depend on $N_s$. Thus, for a given number $N_{\mathrm{new}}$ of unanalyzed samples, the computational cost corresponding to the process of (a) and (b) is at least $K_r N_r N_{\mathrm{new}} + K_L L_1 N_s^2$, where $N_r$ is the minimum reads number over all samples, and $L_1$ is the number of target regions merged over all samples. The first term comes from counting the reads and the second term comes from calculating the distance matrix. Note that $K_r$ is a constant that does not depend on $N_r$ or $N_{\mathrm{new}}$, and $K_L$ is a constant that does not depend on $L_1$ or $N_s$. This form of the computational cost $K_r N_r N_{\mathrm{new}} + K_L L_1 N_s^2$ is the same for case (i) with $N_{\mathrm{new}} = N_s$ and case (ii) with $N_{\mathrm{new}} = 1$, leading to the conclusion that the computational cost of our algorithm is significantly lower than the previous method, especially for case (ii) with sufficiently large $N_s$. We do not have the exact estimate of the coefficients $K_1, K_2, K_r, K_L$, but because $N_r = 3265006 \gg M_{\mathrm{cut}}^*$ and $L_1 = 590650 \gg M_{\mathrm{cut}}^*$ in our case, then $K_r N_r N_{\mathrm{new}} + K_L L_1 N_s^2$ could be costly compared to $K_1 M_{\mathrm{cut}} N_s^2$. In practice, even in case (i) with $N_s = 77$, we numerically found that the computational cost of our algorithm is lower due to our algorithm not using the process of merging reads unlike [7].

## How to relate the best parameters to genomic context

In order to understand why ATAC-seq data under the condition of $(M_{\mathrm{cut}}, p_G) = (64000, 10^{-2})$ was well classified, we analyzed the properties of the peaks with higher rankings.

The result of the previous section suggested that peaks of $\{g_k\}_{k=1}^{M^*_{cut}}$ with $M^*_{cut} = 64000$ included key regions for characterizing cell types. Therefore, we investigated which functional genomic regions such as promoters, enhancers, etc. are dominantly related to these top 64000 peaks.

**Functional annotation of peaks depending on rank.** In order to investigate functional annotations on the genome overlap with ATAC-seq peaks data, we applied the top 80000 peaks in three cell types (HSC, B cells, and Mono) to the 15-state ChromHMMmodel data. One can obtain data of the biological functions on the genome for HSC, B cells, and Mono from an integrative analysis of 111 reference human epigenome datasets, where we used the data of E032 for B cells, E035 for HSC, and E029 for Mono (https://egg2.wustl.edu/roadmap/data/byFileType/chromhmmSegmentations/ChmmModels/coreMarks/jointModel/final/) [17].

ATAC-seq peaks were ranked according to *p*-values and divided into groups consisting of 1000 peaks. Then we calculated the average ratio and the standard deviation for each of the 15 states over all samples in each cell type. For an explicit description, let us introduce a set of functional annotations, $\mathbb{W} := \{\mathbb{W}_y\}_{y=1}^{15}$, where $\mathbb{W}_y$ is the set of regions on the genome, each of which corresponds to functional annotation *y*. We want to know how many peaks, *k*, of every 1000 peaks belong to each functional annotation *y*. For this purpose, we define

$$E^y_x := \{x \le k < x + 1000 \mid \exists(\gamma_k, [\sigma, \epsilon]) \in \mathbb{W}_y \ such \ that \ \sigma \le (\alpha_k + \beta_k)/2 \le \epsilon\},$$

where $g_k = (\gamma_k, \alpha_k, \beta_k)$ is the peak position. We computed $|E^y_x|/1000$ for $x \in \{1 + (j - 1) \times 1000\}_{j=1}^{80}$, as shown in Fig 12. Note that we used the position of the peak center, $(\alpha_k + \beta_k)/2$, to annotate biological function.

As shown in Fig 12, most of the peaks with higher rankings belonged to "Active TSS", which was related to the promoters of active genes, but as the rank went down, the ratio of peaks from enhancer regions started to increase. As the rank went down further, the ratio of peaks from "quiescent-low" regions started to increase. The ratio of peaks from promoters and enhancers crossed at around peak rank 10000 and the ratio of peaks from enhancers and "quiescent-low" regions crossed at around peak rank 60000. Therefore, we concluded that the number around the 64000th peak is strongly related to the point that the contribution of "quiescent-low" regions to the Hamming distances exceeds the contribution of enhancer regions to the Hamming distances.

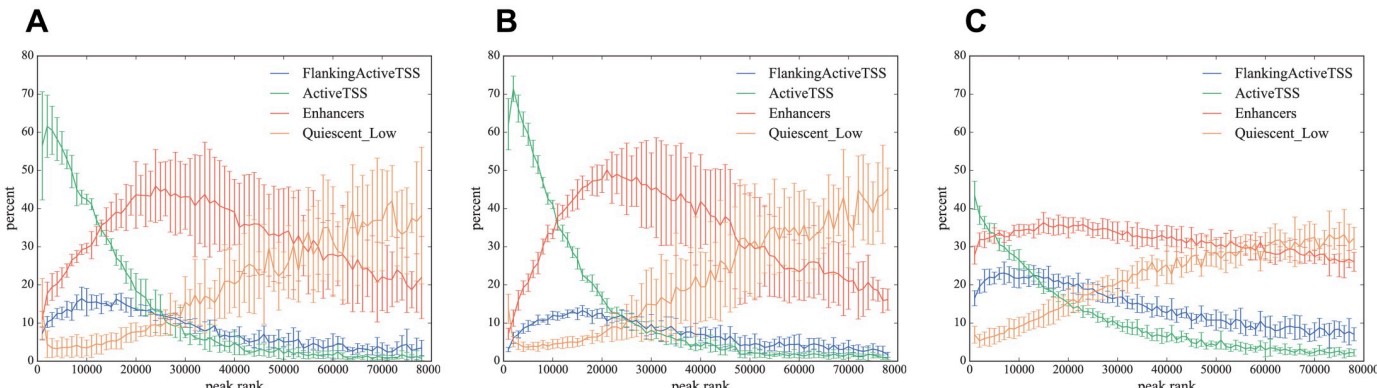

**Fig 12. Functional annotations of peaks.** Percentage ($100 \times |E^y_x|/1000$) of functional annotations in every 1000 peaks for B cells (A), Mono (B), and HSC (C). Only the functional annotations that have maximum percentages $\ge 12\%$, $y \in$ {FlankingActiveTSS, ActiveTSS, Enhancers, Quiescent_Low}, are shown.

Note that the type penalty of HSC under the condition $(M_{\mathrm{cut}}^*, p_G^*)$ was not as good as that of B cells or Mono, and the functional annotation result of HSC did not show clear behaviors compared with B cells and Mono ([Fig 12C]), which may partially explain the worse type penalty of HSC ([Fig 10B]).

## Variations of hierarchical clustering methods

In general, when one performs data clustering, the effect of variations of the clustering algorithms and the effect of loss of data on the clustering output should be considered.

First we considered the dependence of the clustering results on the variations of the clustering algorithms. Besides Ward' method which we used until here, there are several hierarchical clustering methods including UPGMA (Unweighted Pair Group Method with Arithmetic mean), WPGMA (Weighted Pair Group Method with Arithmetic Mean), UPGMC (Centroid Clustering or Unweighted Pair Group Method with Centroid Averaging), and WPGMC (Median Clustering or Weighted Pair Group Method with Centroid Averaging). We performed optimization also with UPGMA, as shown in [Fig 13], and found that the minimum value of the penalty is 36 with $M_{\mathrm{cut}}$ = 12000. The other methods give worse results in general. Specifically, the minimum values of the penalty we found were 59 for WPGMA with $M_{\mathrm{cut}}$ = 20000, 127 for UPGMC with $M_{\mathrm{cut}}$ = 30000, and 149 for WPGMC with $M_{\mathrm{cut}}$ = 35000. These results suggested that Ward's method giving 18 as the minimum value of the penalty was a better choice than that of the other methods for our purpose.

## Robustness of our best clustering against the loss of data

Regarding the loss of data, let us consider making new reads data $\hat{\mathbf{R}}$ from original data $\mathbf{R}$. Specifically, we set $r$ with $0 \le r \le 1$ as the probability of randomly removing $\lceil rN_{\mathrm{r}} \rceil$ reads from $\mathbf{R}$ with the uniform distribution, where $\lceil \chi \rceil$ means the minimum integer larger than or equal to $\chi$. Thus we can obtain $\hat{\mathbf{R}} = \{\mathbf{R}_i'\}_{i=1}^{N_{\mathrm{r}} - \lceil rN_{\mathrm{r}} \rceil}$, where $\mathbf{R}_i'$ is one read in $\mathbf{R}$. Using this procedure, we computed $\lambda$ for $(M_{\mathrm{cut}}^*, p_G^*) = (64000, 10^{-2})$. As shown in [Fig 14B], when ratio $r$ was increased, the value of $\lambda$ was constant until $r = 0.007$ and gradually increased thereafter. In the region

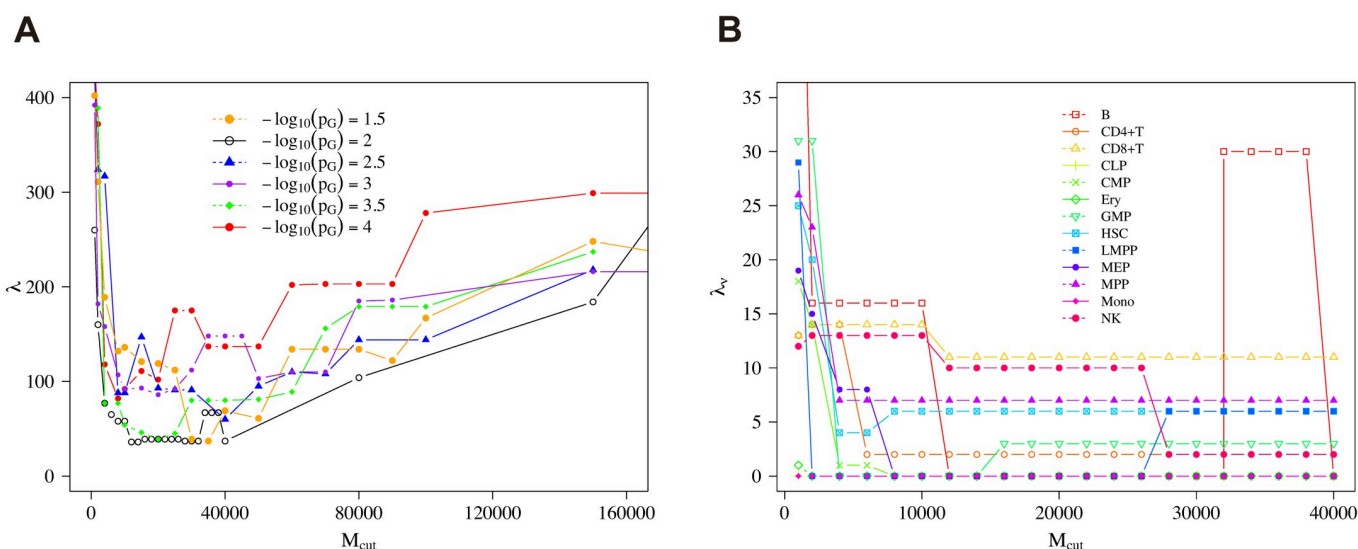

**Fig 13. Penalty by UPGMA method.** The distribution of global penalty $\lambda$ (A) and type penalty $\lambda_v$ for each cell type $v$ (B) along with $M_{\mathrm{cut}}$ with parameter $p_G = 10^{-2}$ by using UPGMA.

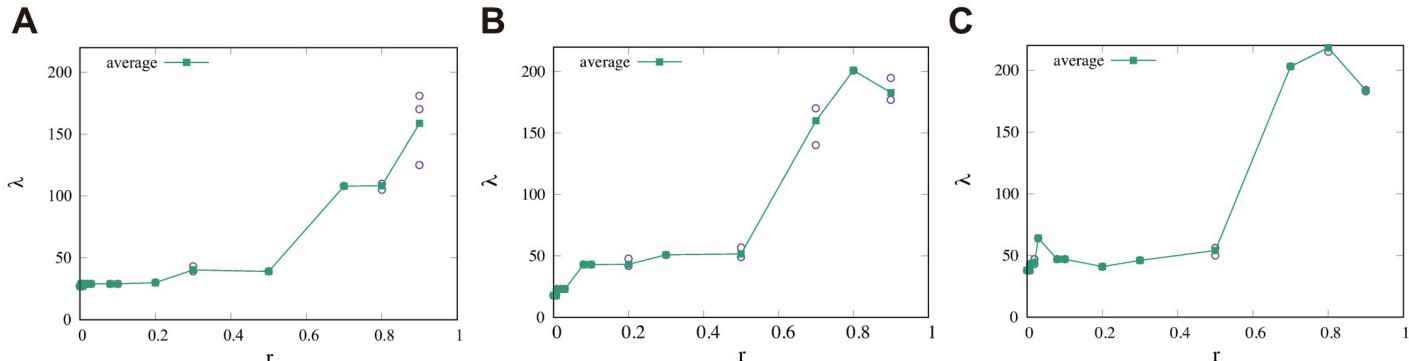

**Fig 14. Robustness of penalty against the loss of reads data.** The effect of the loss of reads on the global penalty λ. Reads were removed randomly from the uniform distribution with probability $r$. Then global penalty λ was calculated with parameter $M_{cut}$ = 30000 (A), $M_{cut}$ = 64000 (B) or $M_{cut}$ = 80000 (C). Each circle indicates one sample and each square indicates the average over samples at the same $r$ value.

$r \geq 0.7$, λ increased dramatically. Note that $r = 0$ gave λ = 18 and the highest possible value of λ for 77 samples is 924. Thus, we concluded that for small $r$, the average penalty tended to be stably close to that of $r = 0$.

Further, we investigated λ for different values of $M_{cut}$ than 64000 to check the robustness of $M_{cut}^*$ against random selections. Specifically, we investigated the behavior of λ by varying $r$ for $M_{cut}$ = 30000 and 80000 with $p_G^* = 10^{-2}$. The minimum value of λ as a function of $r$ was 27 for $M_{cut}$ = 30000 and located at $r = 0$ (Fig 14A) and was 38 for $M_{cut}$ = 80000 and again located at $r = 0$ (Fig 14C). Note that in the region $r \geq 0.08$, λ for $M_{cut}$ = 30000 was smaller than λ for $M_{cut}$ = 64000, which suggested that $M_{cut}^*$ becomes less than 64000 when the data size is decreased.

Thus, for the present data size, we concluded that our algorithm was stable against small losses of the data and it could also work well by adjusting $M_{cut}$ for losses of data up to 50 percent. The obtained results imply that when the given data size is increased, our algorithm becomes more stable or potentially achieves better clustering with a smaller penalty than our current best clustering.

## Application to leukemic cells

To evaluate the practicality of our algorithm with the optimized parameters $(M_{cut}^*, p_G^*)$ on cancer research, we analyzed three types of leukemia: CLL, AML, and ATL, by calculating Ward's distance function, $\mathcal{H}_{Ward}(\zeta, \mathbb{S}_v)$, between a given leukemia sample $\zeta$ and all samples $c \in \mathbb{S}_v$ of cell type $v$. (See Materials and methods for details of $\mathcal{H}_{Ward}$).

To separate normal and leukemic cells effectively, information about the cell surface markers was used. CLL is a disease that is characterized by the clonal proliferation of malignant B lymphocytes. Leukemic cells from CLL patients were purified by using the cell surface markers CD5 and CD19, which are commonly used as markers for CLL (Table 2) [18].

The AML samples analyzed in this study were divided into three stages, preleukemic HSC (pHSC), leukemia stem cells (LSC), and AML blasts by cell surface markers according to [7] (Table 2). Briefly summarizing these three types, HSC that acquired founder mutations become pHSC, which expand to generate preleukemic clones. The subsequent acquisition of progressor mutations creates LSC, which can self-renew and produce AML blasts [19]. It has been reported that mature LSC populations more closely resemble normal GMP, and immature LSC populations are functionally similar to LMPP [20]. A recent study has revealed that CD99-positive cells are almost entirely composed of LMPP-like cells in the sense of Ref. [21].

**Table 2. Immunophenotypes of leukemic samples.** Immunophenotype of CLL [8]: Note that B cells are CD19+, as shown in Table 1. Immunophenotype of AML [7]: SSC-high means that the intensity of side scatter in the flow cytometry is high. Note that HSC, MPP, and LMPP are Lin-, CD34+, CD38- as shown in Table 1. Immunophenotype of ATL [24, 25]: Note that CD4+T cells are CD4+, as shown in Table 1.

| Type of sample | Marker expression |
| --- | --- |
| CLL | CD19+, CD5+ |
| AML pHSC | Lin-, CD34+, CD38-, TIM3-, CD99- |
| AML LSC | Lin-, CD34+, CD38-, TIM3+, CD99+ |
| AML Blast | Non-LSC; CD45-Intermediate, SSC-High |
| ATL | CD4+, CADM1+ |

Thus, the LSC used in our study, which are CD99-positive, can be presumed to be LMPP-like LSC.

Human T-cell leukemia virus type 1 (HTLV-1) is a causative agent of ATL and HTLV-1-associated myelopathy/tropical spastic paraparesis (HAM/TSP) [22]. ATL has been subclassified into four clinical subtypes: acute, lymphoma, chronic, and smoldering. The chronic and smoldering subtypes are considered indolent, while patients with the acute or lymphoma subtype generally have a poor prognosis. HTLV-1 can infect a variety of cell types, but more than 90% of infected cells are CD4+ memory T cells in vivo [23]. In order to specifically separate HTLV-1-infected cells from other normal T-cells, Cell adhesion molecule 1 (CADM1/TSLC1) is used because of its sensitivity and specificity [24, 25]. Thus, in this study, to purify leukemic cells (HTLV-1 infected cells) from the peripheral blood mononuclear cells (PBMC) of ATL patients, we used the cell surface markers shown in Table 2.

The objective of our analysis using leukemic samples was to evaluate which type of hematopoietic cell is closest to a given leukemic sample at the chromatin level. Specifically, we added the ATAC-seq data of a leukemic sample to healthy hematopoietic ATAC-seq data and calculated the Hamming distances where $(M_{\text{cut}}^*, p_G^*) = (64000, 10^{-2})$ is used. We computed $\mathcal{H}_{\text{Ward}}(\zeta, \mathbb{S}_v)$ as the distance between cell type $v \in \mathbb{T}$ and leukemic sample $\zeta$; in this case, sample $\zeta$ was extracted from one patient.

We define the $q$-th closest cell type of sample $\zeta$ as type $v_\zeta^{(q)} \in \mathbb{T}$ to provide the $q$th minimum of $\mathcal{H}_{\text{Ward}}(\zeta, \mathbb{S}_v)$ in terms of $v$. Using this quantity, we define the rank gap between a given reference type $T_0 \in \mathbb{T}$ and sample $\zeta$ as

$$G_{T_0, \zeta} = q - 1,$$

such that $T_0 = v_\zeta^{(q)}$. In particular, we call $v_\zeta^{(1)}$ the closest type of sample $\zeta$. Note that rank gap $G_{T_0, \zeta} = 0$ holds when $T_0 = v_\zeta^{(1)}$. Thus, we not only revealed the closest cell type, but also identified the second, third, and so on closest cell type, and quantified the difference between the characterization results of our algorithm and a given type as the "rank gap".

As shown in Table 3, by calculating the Hamming distance between each CLL sample and a set of hematopoietic cells, we found that the closest cell type for all CLL samples was B cells, which coincides well with the characteristics of CLL cell surface markers. This result led us to conjecture that our method could infer the cell type of a given leukemic cell characterized by immunophenotypes with using only its ATAC-seq data.

In order to assess the applicability of our method to leukemia whose cell of origin is not uniform and has high levels of heterogeneity between cases, we analyzed AML samples [7]. We found that the results of our analysis for pHSC and Blast had substantial overlap with those of

**Table 3. Classification of ATAC-seq data of CLL samples.** "Closest cell type" computed by our algorithm.

| sample name ($\zeta$) | SRR number | Type consistent to surface marker | "closest cell type" calculated by our algorithm ($v_{\zeta}^{(1)}$) |
|---|---|---|---|
| CLL1 | SRR6762820 | B | B |
| CLL2 | SRR6762844 | B | B |
| CLL3 | SRR6762861 | B | B |
| CLL4 | SRR6762895 | B | B |
| CLL5 | SRR6762925 | B | B |
| CLL6 | SRR6762952 | B | B |
| CLL7 | SRR6762968 | B | B |

a previous study [7], where 12 out of 16 samples for pHSC and 13 out of 18 samples for Blast are overlapped, as shown in Table 4. However, in the case of LSC, we found differences between the results of our analysis and those from [7]. Most of the LSC samples were closest to LMPP using our algorithm, but to GMP in [7]. As mentioned above, the LSC used in the present study were CD99-positive and are presumed to be composed of LMPP-like cells, which suggests that our characterization by using information of the Hamming distance infers the cell type with high accuracy, though further investigation is required.

Finally we analyzed ATL samples (See Materials and methods for details of sample preparation). When we calculated the Hamming distance between each ATL sample and a set of hematopoietic cells, we found that the closest cell type for two ATL samples was Mono (hereafter we term these samples "Mono-like ATL"), while that of the other samples was CD4$^+$T, as shown in Table 5. Surprisingly, the two Mono-like ATL samples were categorized into chronic-type ATL. Since CD14 is the marker of Mono (Table 1), we investigated the CD14 gene expression pattern in CD4$^+$T, Mono and ATL samples. Particularly, we calculated the ratio of the CD14 reads count to the CD4 reads count from RNA-seq data and found that the two Mono-like ATL samples exhibited higher values among all ATL samples (Fig 15). In this way, the obtained results led us to conjecture that our algorithm could infer the cell phenotype, potentially including clinical subtypes, only using ATAC-seq data. However, we need to analyze more samples to validate this conclusion.

## Discussion

In this paper, we presented a new algorithm to systematically perform clustering of epigenomic data using the Hamming distance, which enabled us to find optimal parameters of the data reduction toward cell-type classification. This algorithm has one clear advantage in terms of computational cost compared to a previous method using targeted regions merged over samples [7]. Especially, when adding new samples to the analysis, we only have to calculate the distances between newly appearing pairs of samples and not between preexisting samples. The computational cost of the presented systematic algorithm is significantly lower for this situation compared to the previous method with merging targeted regions. Furthermore, this algorithm was found to effectively detect the closest cell type of a leukemic sample, with the results being broadly consistent with the characterization of leukemic samples by cell surface markers or RNA-seq. Thus, the developed algorithm potentially serves as a screening for the phenotype of a leukemia sample by using the ATAC-seq data of the sample as input.

As a next step, we need to investigate if our constructed algorithm is robust for other existing methods and data. For example, for the same data of hematopoietic cells, we replaced the Hamming distance with the Dice coefficient, which has been used in the CODEX project [26]

**Table 4. Classification of ATAC-seq data of AML samples.** Comparison between the "closest normal cell" in Fig 6i of [7] and "closest cell type" computed by our algorithm. The second, the third, and . . .-th "closest type" were also identified by our algorithm. The "rank gap" represents the difference of the result between the two analytical methods. For example, the "closest normal cell" of sample SU351-pHSC is MPP in [7], but is LMPP by our algorithm. "MPP" was the second "closest cell type". Thus, the rank gap was calculated as 2-1 (= 1). If the results from the two analytical methods coincide with each other, the rank gap is 0.

| sample name ($\zeta$) | SRR number | "closest normal cell" ($T_0$) calculated in Fig 6i from Ref. [7] | "closest cell type" calculated by our algorithm ($v_\zeta^{(1)}$) | rank gap ($G_{T_0,\zeta}$) |
|---|---|---|---|---|
| SU654-pHSC | SRR2920595 | MPP | MPP | 0 |
| SU353-pHSC | SRR2920571 | MPP | MPP | 0 |
| SU351-pHSC | SRR2920568 | MPP | LMPP | 1 |
| SU209-pHSC1 | SRR2920564 | GMP | MPP | 4 |
| SU209-pHSC2 | SRR2920562 | GMP | GMP | 0 |
| SU209-pHSC3 | SRR2920561 | GMP | GMP | 0 |
| SU070-pHSC1 | SRR2920557 | HSC | MPP | 1 |
| SU070-pHSC2 | SRR2920556 | HSC | HSC | 0 |
| SU048-pHSC | SRR2920552 | MPP | MPP | 0 |
| SU583-pHSC1 | SRR2920588 | GMP | LMPP | 2 |
| SU583-pHSC2 | SRR2920587 | GMP | GMP | 0 |
| SU575-pHSC | SRR2920584 | MPP | MPP | 0 |
| SU501-pHSC | SRR2920581 | MPP | MPP | 0 |
| SU496-pHSC | SRR2920579 | MPP | MPP | 0 |
| SU484-pHSC | SRR2920576 | MPP | MPP | 0 |
| SU444-pHSC | SRR2920574 | MPP | MPP | 0 |
| SU654-LSC | SRR2920594 | LMPP | LMPP | 0 |
| SU583-LSC | SRR2920586 | GMP | LMPP | 1 |
| SU575-LSC | SRR2920583 | GMP | LMPP | 2 |
| SU496-LSC | SRR2920578 | GMP | GMP | 0 |
| SU444-LSC | SRR2920573 | GMP | LMPP | 1 |
| SU353-LSC | SRR2920570 | GMP | LMPP | 1 |
| SU209-LSC | SRR2920559 | GMP | LMPP | 1 |
| SU070-LSC | SRR2920555 | GMP | LMPP | 1 |
| SU654-Blast | SRR2920593 | GMP | LMPP | 1 |
| SU444-Blast | SRR2920572 | Mono | Mono | 0 |
| SU353-Blast | SRR2920569 | GMP | GMP | 0 |
| SU351-Blast | SRR2920567 | Mono | GMP | 1 |
| SU209-Blast | SRR2920558 | GMP | GMP | 0 |
| SU070-Blast1 | SRR2920554 | Mono | Mono | 0 |
| SU070-Blast2 | SRR2920553 | Mono | Mono | 0 |
| SU048-Blast1 | SRR2920551 | GMP | GMP | 0 |
| SU048-Blast2 | SRR2920550 | GMP | Mono | 1 |
| SU048-Blast3 | SRR2920549 | GMP | GMP | 0 |
| SU048-Blast4 | SRR2920548 | GMP | Mono | 1 |
| SU048-Blast5 | SRR2920547 | GMP | GMP | 0 |
| SU048-Blast6 | SRR2920546 | GMP | GMP | 0 |
| SU583-Blast | SRR2920585 | GMP | GMP | 0 |
| SU575-Blast | SRR2920582 | GMP | LMPP | 1 |
| SU501-Blast | SRR2920580 | Mono | Mono | 0 |
| SU496-Blast | SRR2920577 | GMP | GMP | 0 |
| SU484-Blast | SRR2920575 | Mono | Mono | 0 |

**Table 5. Classification of ATAC-seq data of ATL samples.** Clinical subtypes of ATL samples and "closest cell type" computed by our algorithm.

| sample name ($\zeta$) | DRR number | clinical subtypes | "closest cell type" calculated by our algorithm ($v_\zeta^{(1)}$) |
|---|---|---|---|
| ATL1 | DRR250710 | Acute | CD4$^+$T |
| ATL2 | DRR250711 | Acute | CD4$^+$T |
| ATL3 | DRR250712 | Acute | CD4$^+$T |
| ATL4 | DRR250713 | Acute | CD4$^+$T |
| ATL5 | DRR250714 | Acute | CD4$^+$T |
| ATL6 | DRR250715 | Chronic | Mono |
| ATL7 | DRR250716 | Chronic | Mono |

to quantify the differences between two samples, but found the results with $p_G = 10^{-2}$ were not improved in terms of the penalty. We also compared our algorithm with DiffBind [27], which is commonly used as a ChIP-seq differential analysis tool, but again found that DiffBind with its default setting did not give a better clustering result. Note that there are other existing methods and data to be checked in the future.

A unique point of our constructed algorithm is that we only used ATAC-seq data without gene expression data. Our analysis suggests that ATAC-seq data itself contains enough information to determine cell types even in the absence of regional annotation data such as promoters or enhancers. This feature implies that our algorithm reveals elusive epigenomic properties that significantly affect the phenotype of cell types. Another advantage of our algorithm is that we do not assume a strong property for the statistics of the reads data, which is otherwise implicitly assumed when quantile normalization is performed. Instead of using the strong assumption, we took a data-driven approach for the normalization of the reads data, where we pre-analyzed the statistics of the reads data before performing any normalization.

Finally, our algorithm could extend its application to leukemic samples whose properties are uncertain. We also expect that our whole approach with slight modifications will be applicable to other epigenetic sequencing data such as ChIP-seq and bisulfite sequencing available, for example, from The International Human Epigenome Consortium (https://epigenomesportal.ca/ihec/), ROADMAP Epigenomics (http://www.roadmapepigenomics.org/) and many other resources, whose target regions for the analysis are not uniform between samples.

## Materials and methods

### Ethics statement

Experiments using clinical samples were conducted according to the principles expressed in the Declaration of Helsinki and approved by the Institutional Review Board of Kyoto University (permit numbers G310 and G204). ATL patients provided written informed consent for the collection of samples and subsequent analysis.

### Sequencing sample preparation

ATL patient PBMCs were thawed and washed with PBS containing 0.1% BSA. To discriminate dead cells, we used the LIVE/DEAD Fixable Dead Cell Stain Kit (Invitrogen). For cell surface staining, cells were stained with APC anti-human CD4 (clone: RPA-T4) (BioLegend) and anti-SynCAM (TSLC1/CADM1) mAb-FITC (MBL) antibodies for 30 minutes at 4°C followed by a wash with PBS. HTLV-1 infected cells (CADM1+ and CD4+) were sort-purified with FACS Canto (Beckman Coulter) to reach 98–99% purity. Data was analyzed by FlowJo software

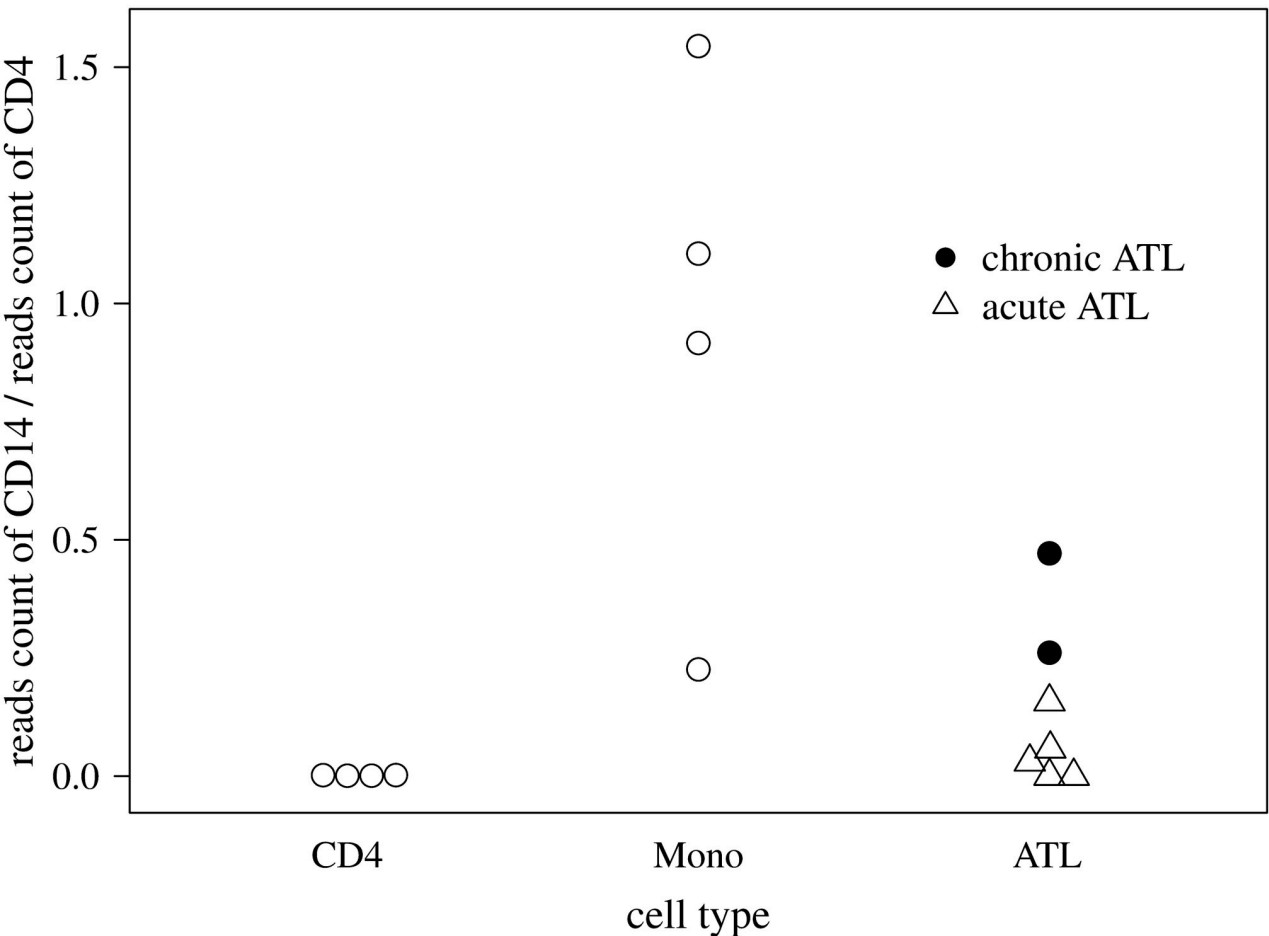

**Fig 15. Comparison of RNA-seq data among CD4+T, Mono and ATL samples.** The reads count of CD14 over the reads count of CD4 from RNA-seq data of CD4+T, Mono, and ATL samples.

(Treestar). Soon after the sorting, 10000-50000 HTLV-1 infected cells were centrifuged and used for ATAC-seq as previously described [5]. Total RNA was isolated from the remaining cells using the RNeasy Mini Kit (Qiagen). Library preparation and high-throughput sequencing were performed by Macrogen Inc. (Seoul, Korea). The diagnostic criteria and classification of clinical subtypes of ATL were performed as previously described [28]. 77 ATAC-seq datasets from 13 human primary blood cell types and datasets from 42 AML patients were obtained from the Gene Expression Omnibus (GEO) with accession number GSE74912 [7]. ATAC-seq datasets from 7 CLL patients were obtained from GSE111015 [18] and RNA-seq datasets of CD4+T and Mono cells were obtained from GSE74246 [7].

## Sequencing data analysis

ATAC-seq reads were aligned using BWA version 0.7.16a [29] with default parameters. SAMtools [30] was used to convert SAM files to compressed BAM files and sort the BAM files by chromosome coordinates. PICARD software (v1.119) (http://broadinstitute.github.io/picard/) was then used to remove PCR duplicates using the MarkDuplicates options. Reads with mapping quality scores less than 30 were removed from the BAM files. For peak calling, MACS2

(v2.1.2) software was used [15]. RNA-seq data were aligned to human reference genome hg19 using STAR 2.6.0c [31] with the - -quantMode GeneCounts function. Normalization was not performed, and only raw reads count data of CD14 and CD4 were used in this study.

## Principles of data reduction

When we analyze preprocessed ATAC-seq data with $\hat{\mathbf{P}}$, we have to care for biases caused by the fact that the amount of reads, $N_r$, depends on the setting of the sample preparation and on the sequencers used. (See S1 Appendix for the explicit construction of $\hat{\mathbf{P}}$.) Normalization is done to remove such biases.

A conventional way to perform normalization is to use quantile normalization, where the distribution of the reads number on certain regions in the DNA is assumed to be the same for all samples [12, 13]. However, there is no strong reason to support this assumption, particularly for sample sets of different cell types. Furthermore, under this assumption, there is a risk that we overlook important differences between different cell types. Therefore, in this paper, we do not assume this property.

An alternative way to perform normalization is to reduce the data into a simple binary value $h_{\gamma,x} \in \{0, 1\}$ on each genomic position $(\gamma, x)$, where $h_{\gamma,x}$ depends on the data size $N_r$ as little as possible. For example, one could determine the state of $h_{\gamma,x} = 1$ and $h_{\gamma,x} = 0$ as an "open" and "closed" chromatin status, respectively, on genomic position $(\gamma, x)$.

In this direction, our ultimate purpose is to look for the "best" principle that determines two states for $h_{\gamma,x}$, by which a set of samples including different cell types are completely classified into groups of the same cell type. We use no information about cell types when determining the value of $h_{\gamma,x}$, because we would like to have an algorithm that can be applied without knowing the cell types.

## Peak-calling with ranking

Currently we do not have the best solution to properly determine two effective states for $h_{\gamma,x}$. As a candidate to approach the best solution, we use the MACS2 algorithm, which was originally invented to analyze ChIP-seq data [15] but is now widely used to estimate the location of open chromatin regions from ATAC-seq data [32, 33].

We would like to find the set of position $(\gamma, x)$ where the number of reads overlapping with position $(\gamma, x)$, $Y_{\gamma,x}(\hat{\mathbf{P}})$, is relatively high in the neighborhood $(\gamma, x)$. The MACS2 algorithm is likely to detect those positions from the data of the reads described by $\hat{\mathbf{P}}$. In our calculation, we use the MACS2 (v2.1.2) callpeak command with option "`--nomodel --nolambda --keep-dup all -p` $p_G$", where we need to set parameter $p_G$ as a parameter of *peak* inference (for details, see [15]).

By applying MACS2 to the input ATAC-seq data, we obtain the following output data structure:

- The label $\gamma_k \in \mathbb{X}$ of the chromosome to which the $k$-th peak has a start position $1 \leq \alpha_k \leq L_\gamma$ and end position $1 \leq \beta_k \leq L_\gamma$ for $1 \leq k \leq M$ (here $M$ is the number of peaks). We call $g_k = (\gamma_k, \alpha_k, \beta_k)$ the *k-th peak region*.

- For each $g_k$, $p$-value $p_k$ with $p_k \leq p_G$ is associated to the $k$-th peak. Note that MACS2 outputs $\log_{10}(1/p_k) = -\log_{10} p_k$ instead of $p_k$.

$\mathbb{X}$ and $L_\gamma$ are the set of all chromosomes and the length of chromosome $\gamma$, respectively (see S1 Appendix for details of the notations). We define $\mathbf{A}$ as

$$\mathbf{A} := (g_k, p_k)_{k=1}^M,$$
$$g_k := (\gamma_k, \alpha_k, \beta_k).$$

By reordering the terms of $k$, we can set $p_k \le p_{k'}$ for any $k < k'$ without loss of information.

In Fig 2, we show the distribution of the peak width $|\beta_k - \alpha_k|$ versus ranking $k$. Note that $g_k$ with high $p_k$ could be affected significantly by the conditions of the experiments including sequencing, because the data above rank value 40000 unnaturally touches the value of the lower limit of width 200, which is predetermined by the MACS2 algorithm. Thus, there is a possibility that peaks with higher $p$-values could strongly depend on both the inference algorithm and the number of reads $N_r$. Those peaks would presumably not contribute to the detection of cell phenotypes. This observation suggests we should remove peaks with higher $p$-values as mentioned in Results.

## Parameterized binarization by cutting off low-ranked peaks

Next we reconsidered how to alleviate biases in the data by introducing threshold number $M_{\text{cut}}$, such that

$$\bar{\mathbf{A}}(M_{\text{cut}}) := \{g_k\}_{k=1}^{M_{\text{cut}}},$$

which leads to the removal of $\{g_k\}_{k=M_{\text{cut}}+1}^M$ as a candidate for the normalization of the ATAC-seq data. Note that $\bar{\mathbf{A}}(M_{\text{cut}} = \infty) = \{g_k \mid (g_k, p_k) \in \mathbf{A}\}$. Then, by using $\bar{\mathbf{A}}$, we may introduce a binary sequence

$$\mathbf{B} := \{h_{\gamma,x}\}_{\gamma \in \mathbb{X}, 1 \le x \le L_\gamma},$$

such that $h_{\gamma,x} = 1$ if there is $k$ satisfying $\alpha_k \le x \le \beta_k$ with $(\alpha_k, \beta_k) \in \bar{\mathbf{A}}$; otherwise $h_{\gamma,x} = 0$ as shown in Fig 4.

$p_G$ and $M_{\text{cut}}$ can be regarded as parameters for determining the value of $h_{\gamma,x}$ within the MACS2 algorithm and what part of the data is taken into account, respectively. Thus, our task under the principle above turns out to be how to determine a proper set of $(M_{\text{cut}}, p_G)$ for the cell-type classification.

## Hamming distance

*The Hamming distance* is often used to compare two binary sequences in information theory (see Section 13 in [14]) and is equal to the number of positions on which two symbols have different values. See Fig 3 for an illustrative explanation.

The Hamming distance between two binary sequences $\mathbf{B}^{c_1}$ and $\mathbf{B}^{c_2}$ with $c_1, c_2 \in \mathbb{S}$ is defined as

$$H(\mathbf{B}^{c_1}, \mathbf{B}^{c_2}) := \sum_{\substack{\gamma \in \mathbb{X} \\ 1 \le x \le L_\gamma}} \delta(h_{\gamma,x}^{c_1}, h_{\gamma,x}^{c_2}),$$

where we define

$$\delta(h_{\gamma,x}^{c_1}, h_{\gamma,x}^{c_2}) = \begin{cases} 1 & (h_{\gamma,x}^{c_1} \ne h_{\gamma,x}^{c_2}) \\ 0 & (h_{\gamma,x}^{c_1} = h_{\gamma,x}^{c_2}). \end{cases}$$

## Algorithm of hierarchical clustering

In this and the next subsection, we recall algorithms for agglomerative hierarchical clusterings and drawing dendrograms. We use two methods, *UPGMA* and *Ward's*. Though they are described in many textbooks (for example, see Chapter 4 in [34]), we need the description in order to define the global penalty and the type penalty. Our description of the algorithms follows [16].

To describe the algorithms, we define two distance functions between two subsets, $\mathbf{C}_1, \mathbf{C}_2 \subset \mathbb{S}$ as follows (for inductive definitions and other distance functions, see Section 4.2 in [34]). One distance function, $\mathcal{H}_{\mathrm{UPGMA}}$ comes from the UPGMA method and is defined as the average of all the distances between samples in $\mathbf{C}_1$ and $\mathbf{C}_2$. Equivalently, we define

$$\mathcal{H}_{\mathrm{UPGMA}}(\mathbf{C}_1, \mathbf{C}_2) := \frac{1}{|\mathbf{C}_1||\mathbf{C}_2|} \sum_{c_1 \in \mathbf{C}_1} \sum_{c_2 \in \mathbf{C}_2} H(\mathbf{B}^{c_1}, \mathbf{B}^{c_2}).$$

If $\mathbf{C}_1$ or $\mathbf{C}_2$ is empty, we set $\mathcal{H}_{\mathrm{UPGMA}}(\mathbf{C}_1, \mathbf{C}_2) = 0$.

Another choice of the distance function, $\mathcal{H}_{\mathrm{Ward}}$, comes from Ward's method and is defined as

$$\mathcal{H}_{\mathrm{Ward}}(\mathbf{C}_1, \mathbf{C}_2) := \sqrt{\frac{D_{1,2}}{|\mathbf{C}_1| + |\mathbf{C}_2|} - \frac{|\mathbf{C}_2|D_1}{|\mathbf{C}_1|(|\mathbf{C}_1| + |\mathbf{C}_2|)} - \frac{|\mathbf{C}_1|D_2}{|\mathbf{C}_2|(|\mathbf{C}_1| + |\mathbf{C}_2|)}}$$

where we define

$$D_1 := \frac{1}{2} \sum_{c_1 \in \mathbf{C}_1} \sum_{c_2 \in \mathbf{C}_1} H(\mathbf{B}^{c_1}, \mathbf{B}^{c_2})^2,$$

$$D_2 := \frac{1}{2} \sum_{c_1 \in \mathbf{C}_2} \sum_{c_2 \in \mathbf{C}_2} H(\mathbf{B}^{c_1}, \mathbf{B}^{c_2})^2,$$

$$D_{1,2} := \sum_{c_1 \in \mathbf{C}_1} \sum_{c_2 \in \mathbf{C}_2} H(\mathbf{B}^{c_1}, \mathbf{B}^{c_2})^2.$$

Again, if $\mathbf{C}_1$ or $\mathbf{C}_2$ is empty, we set $\mathcal{H}_{\mathrm{Ward}}(\mathbf{C}_1, \mathbf{C}_2) = 0$.

In the following, we fix $\mathcal{H}(\mathbf{C}_1, \mathbf{C}_2)$ as $\mathcal{H}_{\mathrm{UPGMA}}$ or $\mathcal{H}_{\mathrm{Ward}}$. We sometimes identify sample $c \in \mathbb{S}$ and subset $\{c\}$ of single element $c$. For example, we write $\mathcal{H}(\mathbf{C}_1, c_2)$ for $\mathcal{H}(\mathbf{C}_1, \{c_2\})$. Note that $\mathcal{H}(\{c_1\}, \{c_2\}) = \mathcal{H}(c_1, c_2) = KH(\mathbf{B}^{c_1}, \mathbf{B}^{c_2})$ where $K = 1$ for $\mathcal{H} = \mathcal{H}_{\mathrm{UPGMA}}$ and $K = 2^{-1/2}$ for $\mathcal{H} = \mathcal{H}_{\mathrm{Ward}}$ by definition.

We define a *cluster* as subset $\mathbb{C}$ of $\mathbb{S}$ with a specified order of elements. Hierarchical clustering is an algorithm that can construct set $\mathbb{M}_{N_s}$ of clusters and order the elements in $\mathbb{S}$ to draw dendrograms.

1. We set $\mathbb{C}_\tau := \{\tau\}$ for $1 \leq \tau \leq N_s$. We do not consider the order of the elements in $\mathbb{C}_\tau$ because they are sets of a single element.

2. We define the list of uncombined clusters as $\mathbb{L}_1 := \{\mathbb{C}_1, \mathbb{C}_2, \ldots, \mathbb{C}_{N_s}\}$ and set the historical list of clusters as $\mathbb{M}_1 = \mathbb{L}_1$.

3. At the $t$-th step $(1 \leq t \leq N_s - 1)$, we define $\mathbb{C}_{t+N_s}, \mathbb{L}_{t+1}$ and $\mathbb{M}_{t+1}$ inductively.

(a) We look up the pair $\mathbb{C}_{\tau'}$ and $\mathbb{C}_{\tau''}$ with $\tau' < \tau''$ in $\mathbb{L}_t$ such that their distance is a minimum; that is,

$$\mathcal{H}(\mathbb{C}_{\tau'}, \mathbb{C}_{\tau''}) = \min_{\substack{\mathbb{C}', \mathbb{C}'' \in \mathbb{L}_t \\ \mathbb{C}' \neq \mathbb{C}''}} \mathcal{H}(\mathbb{C}', \mathbb{C}'').$$

Note that $1 \leq \tau' < \tau'' < t + N_s$ by construction. We consider only the case when the pair is uniquely determined.

(b) We define a new cluster $\mathbb{C}_{t+N_s} = \mathbb{C}_{\tau'} \cup \mathbb{C}_{\tau''}$. If the elements of $\mathbb{C}_{\tau'}$ are ordered as $c_1, c_2, \ldots, c_z$ and the elements of $\mathbb{C}_{\tau''}$ are $c'_1, c'_2, \ldots, c'_{z'}$, then the elements of $\mathbb{C}_{t+N_s}$ are ordered as

$$c_1, c_2, \ldots, c_z, c'_1, c'_2, \ldots, c'_{z'}.$$

(c) We define

$$
\begin{aligned}
\mathbb{L}_{t+1} &:= (\mathbb{L}_t \setminus \{\mathbb{C}_{\tau'}, \mathbb{C}_{\tau''}\}) \cup \{\mathbb{C}_{t+N_s}\}, \\
\mathbb{M}_{t+1} &:= \mathbb{M}_t \cup \{\mathbb{C}_{t+N_s}\}.
\end{aligned}
$$

If $t < N_s - 1$, go to the $(t+1)$-th step.

We can easily see that if we do not consider the ordering, then we have $\mathbb{C}_{2N_s-1} = \mathbb{S}$ as a set. Thus we finally obtain a list of $2N_s - 1$ clusters $\mathbb{M}_{N_s} = \{\mathbb{C}_1, \mathbb{C}_2, \ldots, \mathbb{C}_{2N_s-1}\}$ and an ordering of all elements of $\mathbb{S}$ from $\mathbb{C}_{2N_s-1}$.

## How to draw dendrograms

The *(rooted) dendrogram* displays how our clustering combines pairs of clusters and the distance of the pairs. In the following, we explain an algorithm that introduces new symbols. For details, see [16].

1. If sample $\tau \in \mathbb{S}$ appears in the ordering of $\mathbb{C}_{2N_s-1}$ as the $a_\tau$-th element, then we associate point $n_\tau = (a_\tau, 0)$ in two-dimensional coordinate space to cluster $\mathbb{C}_\tau$. We call point $n_\tau$ the *leaf*, which corresponds to $\mathbb{C}_\tau$.

2. For $1 \leq t \leq N_s - 1$, we inductively associate point $n_{t+N_s}$ to cluster $\mathbb{C}_{t+N_s}$. If $\mathbb{C}_{t+N_s}$ is constructed as the union of $\mathbb{C}_{\tau'}$ and $\mathbb{C}_{\tau''}$ with $1 \leq \tau' < \tau'' < t + N_s$, we associate to $\mathbb{C}_{t+N_s}$ the node

$$n_{t+N_s} = \left(a_{t+N_s} = \frac{a_{\tau'} + a_{\tau''}}{2}, \mathcal{H}(\mathbb{C}_{\tau'}, \mathbb{C}_{\tau''})\right).$$

Note that $\mathbb{C}_{\tau'}$ and $\mathbb{C}_{\tau''}$ are uniquely determined. We call $n_{t+N_s}$ the *node* associated to the $(t + N_s)$-th cluster $\mathbb{C}_{t+N_s}$.

3. We connect $n_{t+N_s}$ with $n_{\tau'}$ and $n_{\tau''}$.

Since each node or leaf $n$ corresponds to cluster $\mathbb{C}$, we can define the *offspring set* $\mathcal{B}_n$ of $n$ as set $\mathbb{C}$ without ordering. Graphically, the offspring set of node $n$ is the set of samples corresponding to leaves branched from node $n$, as displayed in Fig 7. This intuitional explanation is justified, since the $y$-coordinate of the "mother node" $n_{t+N_s}$ is larger than or equal to those of the "child nodes" $n_{\tau'}$, $n_{\tau''}$ if we use Ward's method or UPGMA. Note that there are many

choices to draw dendrograms; for example, at any branching node, we can exchange two branches without any essential change in the data structure.

## Global penalty as a cost function

In this section, we discuss the global penalty, a quantity that measures how the obtained hierarchical clustering differs from our knowledge of cell type classifications. We also give examples displaying the computation of the penalties and extreme situations that represent the theoretical bounds of the penalties. Note that these examples are just for explanation and *not* obtained from actual data.

In our settings, each sample is previously classified by *types*. Explicitly, set $\mathbb{T}$ consists of thirteen types:

$$\mathbb{T} = \{B, CD4^+T, CD8^+T, CLP, CMP, Ery, GMP, HSC, LMPP, MEP, Mono, MPP, NK\}.$$

For each type $v \in \mathbb{T}$, we denote the set of samples classified to type $v$ as $\mathbb{S}_v$. This set could be empty, though it is not in our case. For every pair $v$, $v'$ of distinct types, there are no common elements in $\mathbb{S}_v$ and $\mathbb{S}_{v'}$, and the union of $\mathbb{S}_v$ among all types $v \in \mathbb{T}$ coincides with $\mathbb{S}$. Equivalently,

$$\mathbb{S} = \bigcup_{v \in \mathbb{T}} \mathbb{S}_v.$$

For a given hierarchical clustering constructed in the manner of the previous section, the *type penalty* for type $v$ is the quantity $\lambda_v$ defined as follows. If $\mathbb{S}_v$ is empty, we set $\lambda_v = 0$. Otherwise, since the cluster grows step by step, there is the minimum $\tau$ for $1 \leq \tau \leq 2N_s - 1$ such that $\mathbb{S}_v \subset \mathbb{C}_\tau$. We denote the minimum $\tau$ by $\tau(v)$. Then we define $\lambda_v$ as the number of elements in $\mathbb{C}_{\tau(v)}$ that are not of type $v$. In other words, we set

$$\lambda_v := |\mathbb{C}_{\tau(v)}| - |\mathbb{S}_v|.$$

Since $\mathbb{C}_\tau$ includes all elements of type $v$, we find $\lambda_v \geq 0$. Also since $\mathbb{C}_\tau$ is a subset of $\mathbb{S}$, we find $\lambda_v \leq |\mathbb{S}| - |\mathbb{S}_v|$. Thus we have

$$0 \leq \lambda_v \leq |\mathbb{S}| - |\mathbb{S}_v|.$$

(See Fig 7 for an explanation of type penalties).

For a given hierarchical clustering, the *global penalty* $\lambda$ is defined to be the total sum of type penalties,

$$\lambda := \sum_{v \in \mathbb{T}} \lambda_v.$$

$\lambda$ is bounded as

$$0 \leq \lambda \leq \sum_{v \in \mathbb{T}} (|\mathbb{S}| - |\mathbb{S}_v|) = (|\mathbb{T}| - 1) \cdot |\mathbb{S}|. \tag{1}$$

In our case, since $|\mathbb{T}| = 13$ and $|\mathbb{S}| = 77$, we have $0 \leq \lambda \leq (13 - 1) \cdot 77 = 924$. Note that for a certain class of trees, these upper and lower bounds are not achieved. Fig 16 displays examples of the upper and lower bounds.

Further, we write $\lambda(M_{\text{cut}}, p_G)$ as $\lambda$ to point out that $\lambda$ depends on $(M_{\text{cut}}, p_G)$. Note that $\mathbb{C}_{\tau(v)}$ is equal to $\mathcal{B}_{n_{\tau(v)}}$, which was defined in the previous section.

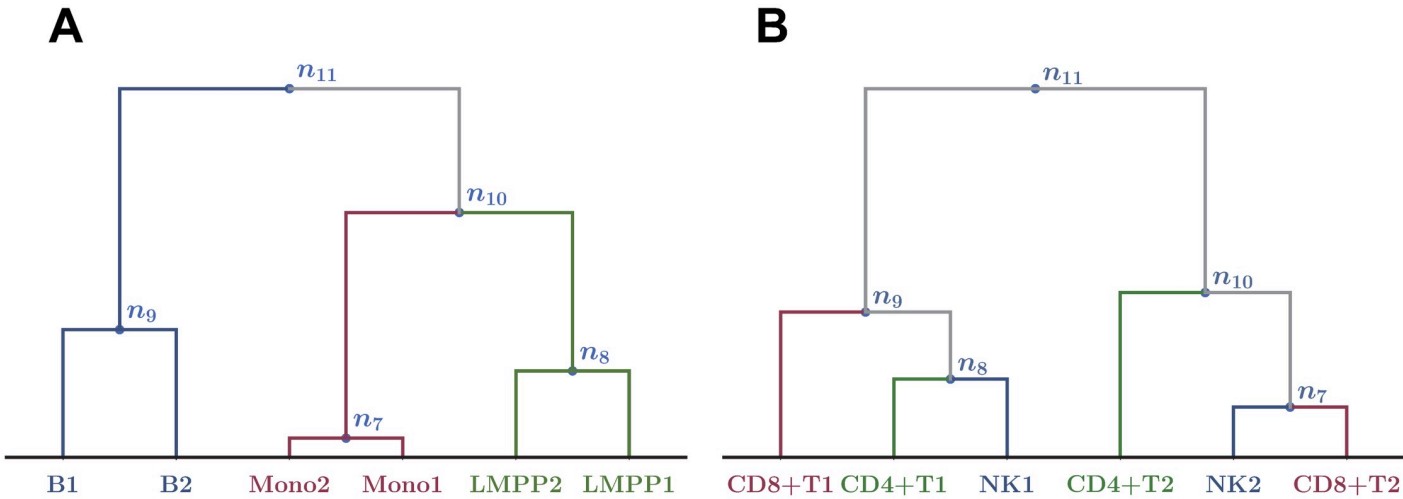

**Fig 16. Examples of dendrograms with extreme penalties.** Note that this dendrogram is constructed using artificial data to explain how to calculate the penalty, though we use the same labels such as Mono1. Both of these dendrograms have six leaves ($|\mathbb{S}| = 6$) that are classified into three types (in these examples, $|\mathbb{T}| = 3$). (A) This example gives the lowest global penalty 0. (B) In this example, we have $\tau(CD4^+T) = \tau(CD8^+T) = \tau(NK) = 11$. Since the corresponding cluster $\mathbb{C}_{11}$ is the whole set $\mathbb{S}$, the local penalty is $6 - 2 = 4$ for each type, and the global penalty is $4 \times 3 = 12$. This result gives the upper bound $(|\mathbb{T}| - 1) \cdot |\mathbb{S}| = (3 - 1) \cdot 6 = 12$ in Eq (1).

## Supporting information

**S1 Appendix. Additional details of sequencing analysis.**
(PDF)

## Acknowledgments

The authors thank P. Karagiannis for valuable comments and proofreading of this manuscript. They also thank MACS Program at Graduate School of Science Kyoto University which allowed this collaboration to be carried out.

## Author Contributions

**Conceptualization:** Azusa Tanaka, Yasuhiro Ishitsuka, Hiroki Ohta, Akihiro Fujimoto.

**Data curation:** Azusa Tanaka.

**Formal analysis:** Azusa Tanaka, Yasuhiro Ishitsuka, Hiroki Ohta.

**Funding acquisition:** Azusa Tanaka, Jun-ichirou Yasunaga, Masao Matsuoka.

**Investigation:** Azusa Tanaka.

**Methodology:** Azusa Tanaka, Yasuhiro Ishitsuka, Hiroki Ohta.

**Project administration:** Azusa Tanaka, Yasuhiro Ishitsuka, Hiroki Ohta, Akihiro Fujimoto.

**Resources:** Azusa Tanaka, Jun-ichirou Yasunaga, Masao Matsuoka.

**Software:** Azusa Tanaka, Yasuhiro Ishitsuka, Hiroki Ohta.

**Supervision:** Azusa Tanaka, Yasuhiro Ishitsuka, Hiroki Ohta, Akihiro Fujimoto.

**Validation:** Azusa Tanaka, Yasuhiro Ishitsuka, Hiroki Ohta.

**Visualization:** Azusa Tanaka, Yasuhiro Ishitsuka, Hiroki Ohta.

**Writing – original draft:** Azusa Tanaka, Yasuhiro Ishitsuka, Hiroki Ohta.

**Writing – review & editing:** Azusa Tanaka, Yasuhiro Ishitsuka, Hiroki Ohta, Akihiro Fuji-
moto, Jun-ichirou Yasunaga, Masao Matsuoka.

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
