## [Decision Letter · Decision Letter 0]

29 Jun 2020

Dear Dr. Tanaka,

Thank you very much for submitting your manuscript "Systematic clustering algorithm for epigenetic data and its application to hematopoietic cells" for consideration at PLOS Computational Biology.

As with all papers reviewed by the journal, your manuscript was reviewed by members of the editorial board and by several independent reviewers. In light of the reviews (below this email), we would like to invite the resubmission of a significantly-revised version that takes into account the reviewers' comments.

We cannot make any decision about publication until we have seen the revised manuscript and your response to the reviewers' comments. Your revised manuscript is also likely to be sent to reviewers for further evaluation.

Sincerely,

Avner Schlessinger

Associate Editor

PLOS Computational Biology

Jason Papin

Editor-in-Chief

PLOS Computational Biology

Reviewer's Responses to Questions

**Comments to the Authors:**

Reviewer #1: Dear Colleagues,

I read with interest your manuscript on identifying cell or tissue type using ATAC-Seq. In it you present a simple algorithm to compute a distance between samples. The text itself is very well written.

Following on the comments in the previous round of reviews, I have two major questions:

- What exactly is the advance over prior art? While it is true that there are few ATAC-Seq specific differential analysis pipelines, a number of ChIP-Seq differential analysis tools such as HOMER, DBChIP and DiffBind are evidently commonly used (1). In fact, a quick search reveals that there are roughly a dozen such methods (2) that can be used to cluster samples based on ChIP-Seq measurements. Have you considered benchmarking your approach against these tools?

- Why is the speed of calculation an issue? Are existing methods too slow? Even assuming a large number of datasets to be compared, computing a matrix of pairwise comparisons is embarrassingly parallel hence the term N_s is not necessarily a concern in practice.

Minor points:

- l85. What does the length of the peaks have to do with data reduction? If you store each peak as a triple (\\gamma, \\alpha, \\beta) (as in l.73), then the difference between \\beta and \\alpha has no incidence on information content.

- l131. Do you have evidence that you algorithm is significantly faster than the approach described in Corces et al? A difference in computational complexity (as derived below) is not a proof of speed, just asymptotic behaviour.

- l151. I don't understand what the comparison of N_r and L_1 to M*_cut brings to the complexity analysis. The Landau notation is about the trend of a function, and the constants within the O(...) are have no bearing to the actual run time, nor to the asymptotic behaviour.

- l173. Are you applying the same M_cut to all samples? Would this not affect results on cell types that have significantly more/fewer accessible regions than others?

Sincerely,

Daniel Zerbino

(1) https://link.springer.com/article/10.1186/s13059-020-1929-3#Fig4

(2) https://hbctraining.github.io/Intro-to-ChIPseq/lessons/08_diffbind_differential_peaks.html

Reviewer #2: Major comments:

1) Some inconsistent concepts are being used throughout the manuscript. In the title, the authors claimed it’s a “clustering algorithm”. However, in the abstract and the main text, the authors descripted it as a ‘data reduction method’. And meanwhile, it was explained as a ‘classification method’. These concepts are all essentially different and mixing them up makes this manuscript read confusing. Especially ‘clustering algorithm’ and ‘classification’ are self-contradictory. Apparently, the authors were utilizing known labels to optimize the loss function or classify samples so it should not be considered as an unsupervised clustering method. The authors need to be careful, consistent, and clear about the method description.

2) For the parameter optimization, instead of using grid search, the authors narrowed down the range of PG first by setting Mcut to infinity and then determine the best parameter pair within the range. But this can be problematic since both parameters contribute to the loss function simultaneously. In other words, the optimal Mcut might fall outside of the range of PG [1.5,4] so the best parameters determined this way might not be optimal, especially when no clear correlation between PG and Mcut was not observed in Figure9 and Figure 10. The authors need to further justify the parameter optimization.

3) Can the determined ‘best parameters’ be generalizable to other cases? It seems different linkage criteria may already result in different solutions of parameters, which means the parameter optimization is a bit sensitive. How to decide the parameters when no labels are given? Also, in the application to leukemic cells, how were the parameters determined?

4) The authors claimed the proposed algorithm works better than quantile normalization however this manuscript lacks a stringent comparison between these two strategies. Line 166-167, although the authors did a basic comparation, comparison details are not described. Are both methods using the same regions? Do the clustering solutions have the same number of clusters? Does the linkage criterion ‘Ward’ give the best performance for quantile normalization? Instead of hierarchical clustering, will another clustering method suit quantile normalization better?

5) How to determine the best parameters for cell-type classification is a critical part for the proposed algorithm. That being so, it makes more sense to put the section of ‘Computational cost of the algorithm’ after “Determination of the best parameters for the best cell-type classification”.

6) Instead of overstating the method as a systematic clustering algorithm for ‘epigenetic data’, it would be better to be more specific since in this manuscript, the algorithm is mainly designed for ATAC-seq and has only been tested on ATAC-seq data.

7) The authors should consider re-organizing the figures and tables (E.g. There is no need to separate table2-4. They can all be merged into one table) to highlight the primary results and improve the readability. Many of them can go to supplementary figures. It would also be very helpful if the authors can add a main figure to illustrate the workflow of the proposed algorithm.

Minor comments:

1) Please define the supersript ‘c’ in Figure3 when first introducing it.

2) Figure7 legend. The correct term for ‘broken line’ is ‘dashed line’

3) Line 164 “Our searching resolution in terms of increasing Mcut was 1000 near Mcut = 64000.” What is ‘1000’ here?

4) Line 287 ‘substantial overlap’. Please give the specific overlap ratio.

**Have all data underlying the figures and results presented in the manuscript been provided?**

Reviewer #1: Yes

Reviewer #2: Yes

PLOS authors have the option to publish the peer review history of their article (what does this mean?). If published, this will include your full peer review and any attached files.

Reviewer #1: **Yes: **Daniel Zerbino

Reviewer #2: No
---

## [Decision Letter · Decision Letter 1]

27 Aug 2020

Dear Dr. Tanaka,

Thank you very much for submitting your manuscript "Systematic clustering algorithm for chromatin accessibility data and its application to hematopoietic cells" for consideration at PLOS Computational Biology.

As with all papers reviewed by the journal, your manuscript was reviewed by members of the editorial board and by several independent reviewers. In light of the reviews (below this email), we would like to invite the resubmission of a significantly-revised version that takes into account the reviewers' comments. Please address the comments by Reviewer 1.

We cannot make any decision about publication until we have seen the revised manuscript and your response to the reviewers' comments. Your revised manuscript is also likely to be sent to reviewers for further evaluation.

Sincerely,

Avner Schlessinger

Associate Editor

PLOS Computational Biology

Jason Papin

Editor-in-Chief

PLOS Computational Biology

Reviewer's Responses to Questions

**Comments to the Authors:**

Reviewer #1: Dear Colleagues,

Thank you for answering most of my questions.

I still do not understand however why your comparison to prior art consists of only one method. Further, the selected competitor was not particularly advanced: the Corces 2016 paper was essentially focused on a new and exciting dataset, not a refined analysis method. I take on board that DiffBind was too slow, but as pointed in my previous review, there are at least a dozen other methods out there.

People have long been doing epigenomic similarity matrices on ChIP-Seq and DNAse-Seq. For example, the website of of the international human epigenome consortium (IHEC) allows you to compute correlation matrices on all their datasets. Also, the CODEX project (1) has been using the Dice coefficient, which is very similar to the Hamming distance used in your method. Why can't these existing methods simply be used on ATAC-Seq data?

Sincerely,

Daniel Zerbino

(1) https://academic.oup.com/nar/article/43/D1/D1117/2439489

Reviewer #2: The authors have sufficiently addressed my concerns. The manuscript has been improved by further clarifying the method and re-organizing the main text structure. I would recommend it for publication.

**Have all data underlying the figures and results presented in the manuscript been provided?**

Reviewer #1: Yes

Reviewer #2: Yes

PLOS authors have the option to publish the peer review history of their article (what does this mean?). If published, this will include your full peer review and any attached files.

Reviewer #1: **Yes: **Daniel Zerbino

Reviewer #2: No
---

## [Decision Letter · Decision Letter 2]

6 Oct 2020

Dear Dr. Tanaka,

We are pleased to inform you that your manuscript 'Systematic clustering algorithm for chromatin accessibility data and its application to hematopoietic cells' has been provisionally accepted for publication in PLOS Computational Biology.

Best regards,

Avner Schlessinger

Associate Editor

PLOS Computational Biology

Jason Papin

Editor-in-Chief

PLOS Computational Biology

Reviewer's Responses to Questions

**Comments to the Authors:**

Reviewer #1: Dear Colleagues,

Thank you for answering my questions.

Best regards,

Daniel

**Have all data underlying the figures and results presented in the manuscript been provided?**

Reviewer #1: Yes

PLOS authors have the option to publish the peer review history of their article (what does this mean?). If published, this will include your full peer review and any attached files.

Reviewer #1: **Yes: **Daniel Zerbino

---

## [Editor Report · Acceptance letter]

24 Nov 2020

PCOMPBIOL-D-20-00774R2 

Systematic clustering algorithm for chromatin accessibility data and its application to hematopoietic cells

Dear Dr Tanaka,

I am pleased to inform you that your manuscript has been formally accepted for publication in PLOS Computational Biology. Your manuscript is now with our production department and you will be notified of the publication date in due course.

With kind regards,

Nicola Davies
